# Characterizing precipitation and soil moisture drydowns in Finland using SMAP satellite data

Kerttu Kouki[1], Andreas Colliander[2]

[1]Finnish Meteorological Institute, P.O. Box 503, 00101 Helsinki, Finland

[2]Jet Propulsion Laboratory, California Institute of Technology, Pasadena, CA 91109, USA

*Correspondence to*: Kerttu Kouki (kerttu.kouki@fmi.fi)

**Abstract.** Precipitation (P) and soil moisture (SM) are critical components of the global water, energy, and biogeochemical cycles, yet their patterns and interrelations in the Arctic are poorly understood. Due to the sparse in situ measurement network, satellites are the only way to observe P and SM in high-latitude regions. This study uses NASA's SMAP satellite to

analyze the relationship between SM and P, assess the feasibility of estimating P from SM, and examine SM drydown patterns in Finland from April to September over 2018-2019. The analysis reveals a notable spatial and temporal variability in SM, with a weaker correlation between P and SM in spring due to snowmelt and a stronger relationship in summer and fall. Water bodies complicate the SM retrieval causing the SM retrievals to saturate. Using the SM2RAIN algorithm, we estimated P from SM data. The algorithm shows promising results, detecting the area of rainfall accurately in most cases but

estimating the intensity of the rainfall is more challenging, particularly for light and very heavy rain. We analyzed SM drydown patterns by fitting an exponential model to each SM drydown period, from which we estimated the exponential decay time scale ($\tau$) and the lower bound of SM ($SM_{min}$). $\tau$ does not show much spatial or temporal variability. The distribution of $\tau$ is positively skewed, with a mode of 1.6 days and a median of 4.0 days, consistent with other studies. The distribution of $SM_{min}$ is also positively skewed, with a mode of 0.14 $m^3$ $m^{-3}$ and a median of 0.17 $m^3$ $m^{-3}$. $SM_{min}$ exhibits

another lower peak at 0.02 $m^3$ $m^{-3}$, the lower limit of SMAP SM retrievals, possibly causing an artifact in the results. $SM_{min}$ shows spatial variability, with the lower bound being slightly higher near water bodies but also showing a more prominent peak at 0.02 $m^3$ $m^{-3}$. Grid cells with dense vegetation and low vegetation agree better with each other, indicating that water bodies particularly affect and complicate SM retrieval. The promising results suggest that the method could be applied across the entire Arctic region.

## 1 Introduction

Precipitation (P) and soil moisture (SM) are critical components of the climate system (Trenberth and Asrar, 2014), both categorized as Essential Climate Variables (ECV) by the Global Climate Observing System (GCOS; Bojinski et al., 2014). P and SM are tightly linked and play vital roles in the global water, energy, and biogeochemical cycles (McColl et al., 2017a; Seneviratne et al., 2010). They affect surface energy fluxes, the carbon cycle, vegetation growth, and land-atmosphere

interactions, and greatly influence the development of various natural hazards, such as droughts, floods, and heat waves (D'Odorico et al., 2003; Entekhabi et al., 1992; Mao et al., 2015; McColl et al., 2017a; Lorenz et al., 2010; Schwingshackl et al., 2017; Trenberth and Asrar, 2014). Accurate estimates of P and SM are essential for climate studies, weather forecasting, and various climatological and hydrological applications (Herold et al., 2016; Wake, 2013; Forootan et al., 2019; Abera et al., 2017). Furthermore, key variables related to SM drydown patterns, including the exponential decay time scale and the lower bound of SM, are crucial inputs for climate models (McColl et al., 2017b; Sellers et al., 1997).

In the Arctic, precipitation occurs in both solid and liquid forms, with distinct seasonal patterns. During winter season, snowfall accounts for over 80% of total precipitation, while rainfall dominates during summer, accounting for more than 70% of the seasonal total (McCrystall et al., 2021). However, due to global warming, a significant decline in the snowfall ratio has already been observed, and projections indicate a continued decrease. By the end of the 21st century, the Arctic is expected to transition from a snow-dominated to a rain-dominated precipitation regime (McCrystall et al., 2021). Additionally, research indicates that total precipitation in the Arctic has increased by 9% during 1971-2019, primarily driven by a 25% rise in rainfall (AMAP, 2021). Future climate projections suggest that precipitation rates will continue to increase (Bintanja and Selten, 2014; McCrystall et al., 2021; Vihma et al., 2015; AMAP, 2021), and research indicates that increased evaporation may substantially contribute to the anticipated rise in P in the Arctic (Bintanja and Selten, 2014).

Despite their importance, P and SM drydown patterns in the Arctic remain poorly understood. Previous satellite-based studies exploring the relationship between P and SM and drydown patterns have often excluded the Arctic region due to the presence of dense forests, water bodies, and seasonally frozen ground, which complicate the SM retrieval (McColl et al., 2017b; Akbar et al., 2018; Sehler et al., 2019). Furthermore, existing P data products exhibit notable variability in the magnitude, frequency, and phase of precipitation in the Arctic (Barrett et al., 2020; Behrangi et al., 2016; Boisvert et al., 2018; Sun et al., 2018). Therefore, it is essential to improve our understanding of the patterns of P and SM in the Arctic, as well as the complex relationship between these two variables.

Due to the sparse in situ measurement network, satellites are usually the only way to observe P and SM in high-latitude regions. Currently, direct satellite-based P estimates lack consistency and sufficient spatiotemporal coverage (Hou et al., 2014; Brocca et al., 2019). However, P can be estimated indirectly using satellite-based SM data (Brocca et al., 2013): SM increases during rainfall and decreases during dry periods. The satellite-based SM measurements provide good spatiotemporal coverage, and unlike direct rainfall measurements, SM data also provide insights into the relationship between SM and P, as well as the dynamics of water after it reaches the ground. The indirect approach has been examined in several global and regional studies using a range of satellite-based SM datasets (e.g., Brocca et al., 2014; Ciabatta et al., 2018; Koster et al., 2016; Mossafa et al., 2023; Miao et al., 2023a). While these studies highlight the potential of this approach, they often exclude the Arctic region due to the challenges associated with SM retrieval (Ciabatta et al., 2018;

Koster et al., 2016; Zhang et al., 2019). Brocca et al. (2019) included the Arctic in their SM2RAIN analysis using data from
the Advanced Scatterometer (ASCAT). However, research has shown that the Soil Moisture Active Passive (SMAP) mission
outperforms ASCAT in high-latitude environments (Chen et al., 2018; Zhang et al., 2020), highlighting the importance of
evaluating whether SMAP can yield more accurate P estimates in these regions. Moreover, satellite-based research on
drydown patterns has generally lacked coverage of the Arctic, creating a need to include this region in future studies.

Soil moisture can be observed using passive microwave satellites operating at low frequencies (Kerr et al., 2010; Entekhabi
et al., 2014). At these low frequencies (<5 GHz), there is a substantial difference in the dielectric constant between water and
dry soil, which affects the microwave emissions (thermal electromagnetic radiation) from the ground. This difference makes
the method sensitive to soil water content (Njoku and Entekhabi, 1996). Additionally, at low frequencies, the influence of the
atmosphere and vegetation is reduced compared to higher frequencies, such as C- and X-bands, which operate at 6 GHz and
11 GHz (e.g., Schmugge et al., 1986). Typically, the low-frequency satellites operate on L-band (1.4 GHz), making them
highly sensitive to the SM in the top layer (0-5 cm) of the surface (Njoku and Entekhabi, 1996). In recent years, several
satellites have been monitoring surface SM, including the National Aeronautics and Space Administration's (NASA) SMAP
satellite, and the European Space Agency's (ESA) Soil Moisture and Ocean Salinity (SMOS) satellite (Entekhabi et al.,
2014; Kerr et al., 2010).


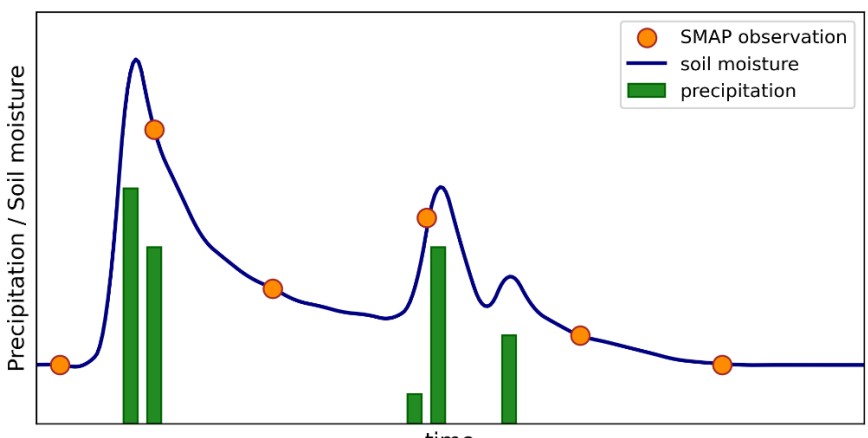

**Figure 1. Schematic example of the relationship between P and SM and satellite-based SM retrievals.**

These satellites perform well and provide generally accurate estimates of surface SM (e.g., Colliander et al., 2022; 2023;
Chan et al., 2016; Chen et al., 2018). However, satellite-based SM retrieval also faces challenges. Firstly, the spatial
resolution is relatively coarse (~40 km), making it difficult to detect small-scale phenomena. Additionally, the revisit time of
SMAP over a specific area typically ranges between 1 to 3 days (Entekhabi et al., 2014), depending on the latitude, which

may complicate the detection of rapid changes in SM (e.g., McColl et al., 2017b). Since rainfall events can be both small-scale and move rapidly, this may lead to inaccuracies in capturing P from SM. For high-latitude areas investigated in this study, the revisit time is about one day, which helps to capture most of the SM minimums and maximums. Figure 1 illustrates a schematic example of the relationship between P and SM along with satellite-based SM retrievals. SM increases during a rainfall event and decreases exponentially during dry periods. Satellites are limited to detecting only part of this temporal variability due to their revisit times. However, despite the difficulties posed by the Arctic environment, the coarse resolution, and non-ideal revisit times, satellites are essential for estimating P and SM in the Arctic due to the lack of in situ measurements.

Due to the reasons described above, it is essential to better understand the relationship between P and SM and to improve our understanding of P and SM patterns across the Arctic. Therefore, this study aims to (1) analyze the relationship between SM and P, (2) investigate whether satellite-based SM data can be used to estimate P rates, and (3) examine SM drydown patterns across Finland. This study is based on SM data from NASA's SMAP satellite and uses ground-based weather radar data and in situ SM and P measurements as reference data. While this study focuses on Finland, the method aims to be applied across the entire Arctic region.

## 2 Data and Methods

The data of this study consist of satellite-based SM data, ground-based weather radar data, and in situ measurements. Table 1 lists the used datasets, and the following sections describe them in more detail.

### 2.1 SMAP SM data

NASA's SMAP satellite has been measuring SM since 2015 (Chan et al., 2016; Entekhabi et al., 2014; Das et al., 2018). The satellite uses an L-band frequency (1.41 GHz), which makes it highly sensitive to the SM in the top layer (0-5 cm) of the surface (Schmugge et al., 1986; Njoku and Entekhabi, 1996). SMAP provides global SM measurements every 2 to 3 days. However, due to Finland's northern location between 60 °N and 70 °N, SM is measured at least daily in northern Finland and every 1 to 2 days in southern Finland (Entekhabi et al., 2014).

We used the enhanced Level 3 (L3) SM product (Version 6) in Polar grid (O'Neill et al., 2021a). Currently, the dual-channel algorithm (DCA) is the baseline algorithm, while earlier versions used the Single Channel Algorithm-Vertical Polarization (SCA-V) as the baseline (e.g., Chan et al., 2018). Our analysis primarily relied on the DCA data, but we also included some additional insights from SCA-V and the Single Channel Algorithm-Horizontal Polarization (SCA-H). We only used the descending, i.e. the 6 am, SM observations. This increases stability, consistency, and retrieval quality due to minor but systematic biases between the ascending and descending retrievals (Colliander et al., 2022). In addition to SM data, we also

used information on surface conditions provided in the product, including quality retrieval flags and surface flags. The SMAP SM retrievals are relatively little affected by rain (Colliander et al., 2020a), so no additional screening was done for retrievals obtained during rain detected with the weather radar or in situ measurements. The native spatial resolution of the SMAP data is 36 km, which is oversampled into a 9 km grid (O'Neill et al., 2021a). We did not perform any spatial processing to the SMAP data.

**Table 1. Datasets used in this study.**

| Dataset | Variable | Resolution / location | Reference |
|---|---|---|---|
| SMAP | SM | 9 km × 9 km, 1 to 3 days | O'Neill et al. (2021a) |
| Ground-based weather radar | P | 250 m × 250 m, hourly | Saltikoff et al. (2010) |
| In situ | SM | Sodankylä, hourly | Ikonen et al. (2016; 2018) |
| | | Saariselkä, hourly | |
| | | Kenttärova, hourly | Nousu et al. (2024) |
| | | Värriö, hourly | SmartSMEAR (2024) |
| | | Hyytiälä, hourly | |
| | P | Sodankylä, hourly | FMI (2024b) |
| | | Saariselkä, hourly | |
| | | Kenttärova, hourly | |
| | | Värriö, hourly | |
| | | Hyytiälä, hourly | |
| SoildGrids 2.0 | Clay fraction Silt fraction Sand fraction Soil organic carbon content Bulk density | 250 m | Poggio et al. (2021) |

**2.2 Radar data**

The Finnish Meteorological Institute (FMI) radar network consists of twelve dual-polarization C-band Doppler radars covering almost the entire country (FMI, 2024a; Saltikoff et al., 2010). In this study, we used radar composite data from eleven of these radars (Fig. 2a), as the most recent radar was installed after the completion of this analysis. The radar composite data have undergone signal processing to remove stationary objects, filter out weak signals, and adjust for non-

meteorological echoes (Saltikoff et al., 2010). The post-processing also included distance correction for rain measurements and the conversion of radar reflectivity to rainfall intensity. We used hourly P rate data to calculate 24-hour P rates based on the SMAP overpass times. The data have a native spatial resolution of 250 meters, and we averaged the data to match the effective SMAP footprint size.

## 2.3 In situ measurements

We used in situ measurements of SM and P from five locations (Fig. 2a). The uneven distribution of these locations results from the limited availability of in situ SM measurements. FMI conducts SM measurements in Sodankylä and Saariselkä (Ikonen et al., 2016; 2018), located in Northern Finland. The measurement network in Sodankylä includes 14 measurements within a single SMAP grid cell, while Saariselkä has 4 measurements, also within one SMAP grid cell. We used a weighted average of the in situ measurements for both locations. The Station for Measuring Ecosystem-Atmosphere Relations (SMEAR) network (Hari and Kulmala, 2005) provides SM measurements in Hyytiälä (SMEAR II) and Värriö (SMEAR I). In Hyytiälä, SM is measured at two sites—one in a forest and the other in a fen. The fifth location for SM measurements is in Kenttärova in Northern Finland (Nousu et al., 2024). In addition to SM data, we included P in situ measurements in our analysis for these five locations where SM data were available. The P data are provided by FMI (FMI, 2024b), and we selected the nearest P measurement location for each SM measurement site.

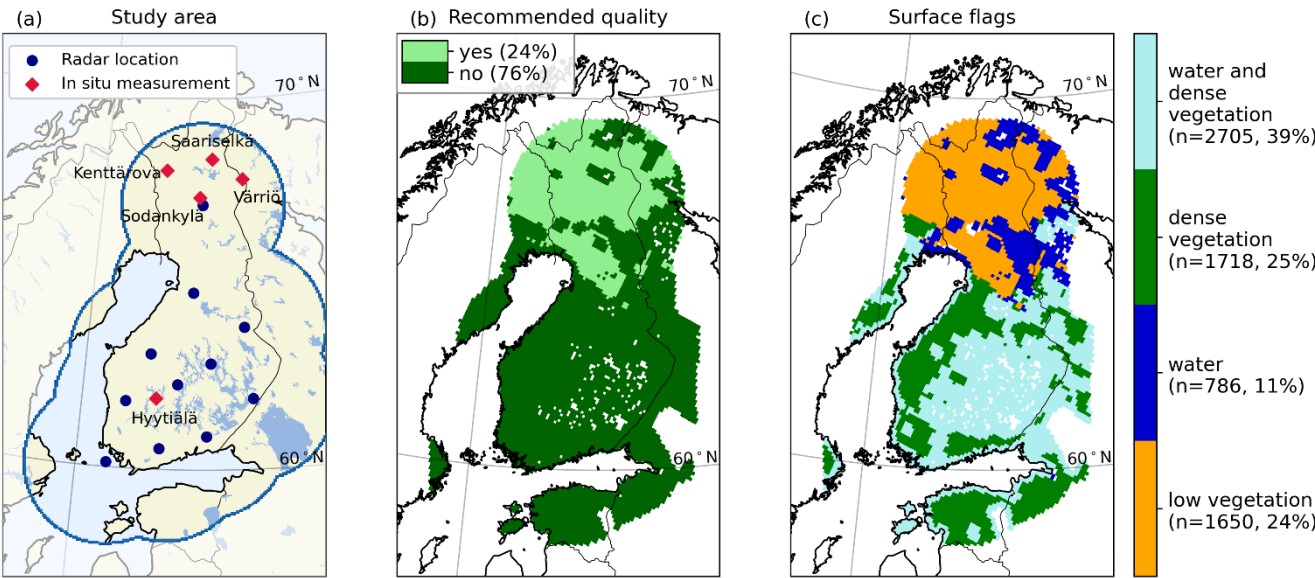

**Figure 2. (a) The study area and the locations of the ground-based weather radars and in situ measurements. (b) The quality retrieval flags for SMAP. (c) The surface flags for SMAP. The percentages indicate the fraction of grid cells included in the category.**

**2.4 Soil properties**

To study the influence of soil properties on SM dynamics, we used data from the SoilGrids 2.0 dataset (Poggio et al., 2021), which also serves as an input for SMAP SM retrievals (O'Neill et al., 2021b). The dataset employs advanced machine learning techniques to model global soil properties at a 250 m spatial resolution, based on approximately 240,000 in situ soil measurements and over 400 environmental covariates. From this dataset, we used key soil texture variables, including sand, silt, and clay fractions, as well as soil organic carbon content and bulk density. Additionally, we derived soil porosity using
the standard relationship:

$$\text{soil porosity} = 1 - \frac{\text{bulk density}}{2.65} \tag{1}$$

where 2.65 g cm$^{-3}$ is the assumed particle density of mineral soil. The spatial distributions of the used variables are shown in
Fig. S1 in the Supplementary material.

**2.5 Methods**

**2.5.1 Study area and study period**

The study area was determined by the coverage of the FMI radar network (Fig. 2a). In the Finnish climate, radar can detect
summer precipitation up to a distance of 250 km (FMI, 2024a). As a result, the study area encompasses nearly all of Finland and parts of neighboring countries. Finland is a challenging region for satellite-based SM retrieval, as large areas of the study region are covered with either water, dense vegetation, or both (Chan et al., 2018). As a result, only 24% of the grid cells in the study area have the recommended quality (Fig. 2b). However, despite the quality concerns, we chose to include all grid cells in our analysis, regardless of the quality retrieval flags, to evaluate how well the methods used in this study perform
under suboptimal SM retrieval conditions. Since the surface conditions affect the SM retrieval, we identified areas covered by water or dense vegetation using the SMAP surface and quality retrieval flags and divided the study area into four classes (Fig. 2c). A grid cell is marked with dense vegetation, if the vegetation water content (VWC) exceeds 5.0 kg m$^{-2}$ and the retrieval is skipped if VWC exceeds 30.0 kg m$^{-2}$ (O'Neill et al., 2021a). Similarly, a grid cell is marked with water if the water fraction exceeds 5% based on Moderate Resolution Imaging Spectroradiometer (MODIS) MOD44W v006 product
(O'Neill et al., 2021a). If the water fraction exceeds 50%, the SM retrieval is skipped. Finland is characterized by an abundance of lakes, but Finnish lakes are often small and vary in shape and size. Consequently, even though many grid cells are flagged for high water fraction, SM is still retrieved in most of them because the water fraction does not exceed 50%.

Finland experiences both maritime and continental climates and is predominantly covered by boreal forests. The mean annual P in Finland ranges mostly between 500 mm and 650 mm (Jokinen et al., 2021). The highest P rates are typically found in Southern and Eastern Finland, while Northern Finland is the driest region. March is the driest month, with precipitation gradually increasing through the summer and peaking in the fall, before declining again afterward. Rainfall is typically irregular, and during the summer, it often occurs as small-scale, localized showers. The first snowfall typically occurs in October or November, depending on the location, and most of Finland remains snow-covered until the end of March. Southern and Central Finland generally become snow-free by the end of April, while in Northern Finland, snow typically melts completely by the end of May (Jokinen et al., 2021). Consequently, the selected study period is from April to September over two consecutive years (2018 and 2019), when the region is predominantly snow-free, but still allowing for the examination of snowmelt's impact on SM.

The year 2018 was characterized by anomalously warm and dry conditions relative to the long-term average (1991-2020). Notably, May and June exhibited monthly mean temperatures 3–5 °C above normal, and precipitation levels approximately 50% below the typical rates. In contrast, 2019 experienced more typical temperatures, with only minor spatial and temporal variations. However, precipitation patterns in 2019 were more variable: April and July recorded only 20–40% the usual rainfall, while May was notably wetter, with precipitation levels reaching up to twice the normal amount.

**2.5.2 SM2RAIN algorithm**

We used the SM2RAIN algorithm to estimate P rates from SM data. The main advantage of the SM2RAIN algorithm is that it requires only SM information to estimate precipitation. This feature is especially beneficial for estimating P in remote Arctic regions, where observational data is often scarce. In the SM2RAIN algorithm, the soil is assumed to act as a natural rain gauge, and changes in SM can be used to estimate rainfall. The algorithm is based on the inversion of the soil water balance equation (Brocca et al., 2013):

$$Z \frac{ds(t)}{dt} = P(t) - R(t) - E(t) - G(t) , \qquad (2)$$

where Z is the soil water capacity, s(t) is the relative SM, t is the time, and P(t), R(t), E(t), and G(t) are the precipitation, surface runoff, evapotranspiration, and drainage (deep percolation and subsurface runoff) rates. Brocca et al. (2015) studied the impact of various terms on P estimates derived from the SM2RAIN algorithm. Their research demonstrated that SM variations and the drainage term are the most significant contributors, accounting for over 90% of the simulated P estimates. The study concluded that neglecting evapotranspiration and surface runoff terms does not weaken the algorithm's performance. Additionally, the use of satellite data with relatively coarse resolution further reduces the impact of surface runoff. The portion that does not infiltrate, due to factors such as impervious land cover or soil, may re-infiltrate downstream

within the scale of a SMAP grid cell (Brocca et al., 2015; 2019). Therefore, during rainfall events, evapotranspiration and surface runoff are assumed to be negligible (Brocca et al., 2014; Kirchner, 2009). The drainage rate can be estimated using the following equation (Brocca et al., 2013; Famiglietti and Wood, 1994):

$\quad G(t) = as(t)^b$ , (3)

where a and b are two parameters expressing the nonlinearity between drainage rate and soil saturation. When rearranging the terms in Eq. (2) and using Eq. (3), we can estimate P using the following equation:

$\quad P(t) = Z\frac{ds(t)}{dt} + as(t)^b$ , (4)

Using the equation above, we can estimate the P rate if we know the relative SM, the changes in relative SM, and three parameters (Z, a, and b).

Before using SMAP SM observations as input for the algorithm, the data needed to undergo a few preprocessing steps. First, in case of missing SMAP data, linear interpolation was used to generate daily values, with a maximum allowable data gap of three days for interpolation. Second, the SMAP data were rescaled between 0 and 1, as relative SM is used as input in the SM2RAIN algorithm. Third, to address high-frequency fluctuations in satellite-based SM observations caused by measurement and retrieval errors, the SMAP data were temporally filtered using the exponential filter approach (Wagner et al., 1999). After these preprocessing steps, the SMAP data were ready to be applied to the SM2RAIN algorithm. The parameters Z, a, and b were calibrated by minimizing the root-mean-square error between the simulated and radar-based 5-day accumulated rainfall. We used data from year 2018 for calibration and data from year 2019 for validation. Since snowmelt affects SM and may lead to erroneous P estimates, we excluded the snowmelt season by only including the months from June to September in the SM2RAIN analysis.


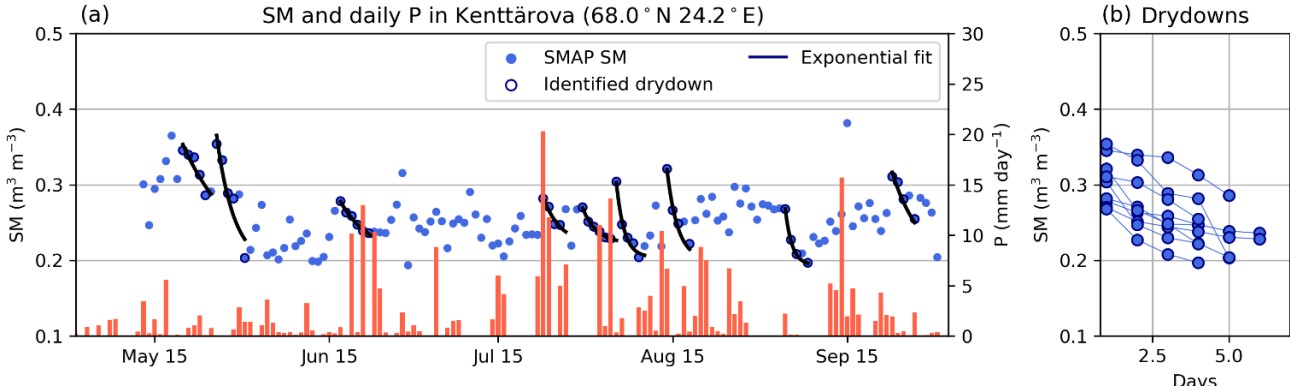

**Figure 3. Timeseries of in situ SM and radar-based P measurements and identified drydowns in Kenttärova.**

### 2.5.3 Drydowns

Drydown periods occur when there is a consistent negative change in SM over time. We identified these drydown periods
separately for each SMAP grid cell. Figure 3 illustrates an example of SMAP SM observations along with the identified
drydowns in Kenttärova. Following the approach of McColl et al. (2017b), we excluded positive increments of less than 5%
of the observed SM range at the site, which helped ensure that noise in the observations did not truncate the drydowns.

Drydown periods can be identified using either SM data alone or by including P time series data. While McColl et al.
(2017b) relied solely on SM information, e.g. Akbar et al. (2018) used both SM and P data, including only days with less
than 1 mm day⁻¹ of P in their analysis. In this study, we employed both methods. We identified drydowns based on SM data
alone and also used radar-based P data with three different thresholds: 0.3 mm day⁻¹, 1 mm day⁻¹, and 4.5 mm day⁻¹. These
thresholds were selected according to FMI's definitions for light, moderate, and heavy rainfall (FMI, 2024c). We analyzed
whether including P data affects the identification of drydown periods.

After identifying the drydowns, we fitted an exponential model (McColl et al., 2017b; Rondinelli et al., 2015; Shellito et al.,
2016) for each drydown that included at least four SMAP observations:

$$SM(t) = \Delta SM \cdot e^{-\frac{t}{\tau}} + SM^*_{min} , \tag{5}$$

where SM(t) is the soil moisture, $\Delta SM$ is the positive increment in SM preceding the drydown, $\tau$ is the estimated drydown
exponential decay time scale, i.e. the time constant, and $SM^*_{min}$ is the estimated lower bound of SM. Following McColl et al.
(2017b), we constrain $SM^*_{min}$ to be less than the lowest SM observed during the drydown, and greater than or equal to the
lowest SM observed over the entire observation period in that grid cell. Here, we note that $SM^*_{min}$ refers to the value

estimated using Eq. (5), while SM$_{min}$ refers to the minimum values observed by SMAP. An example of the exponential fits in Kenttärova is illustrated in Fig 3.

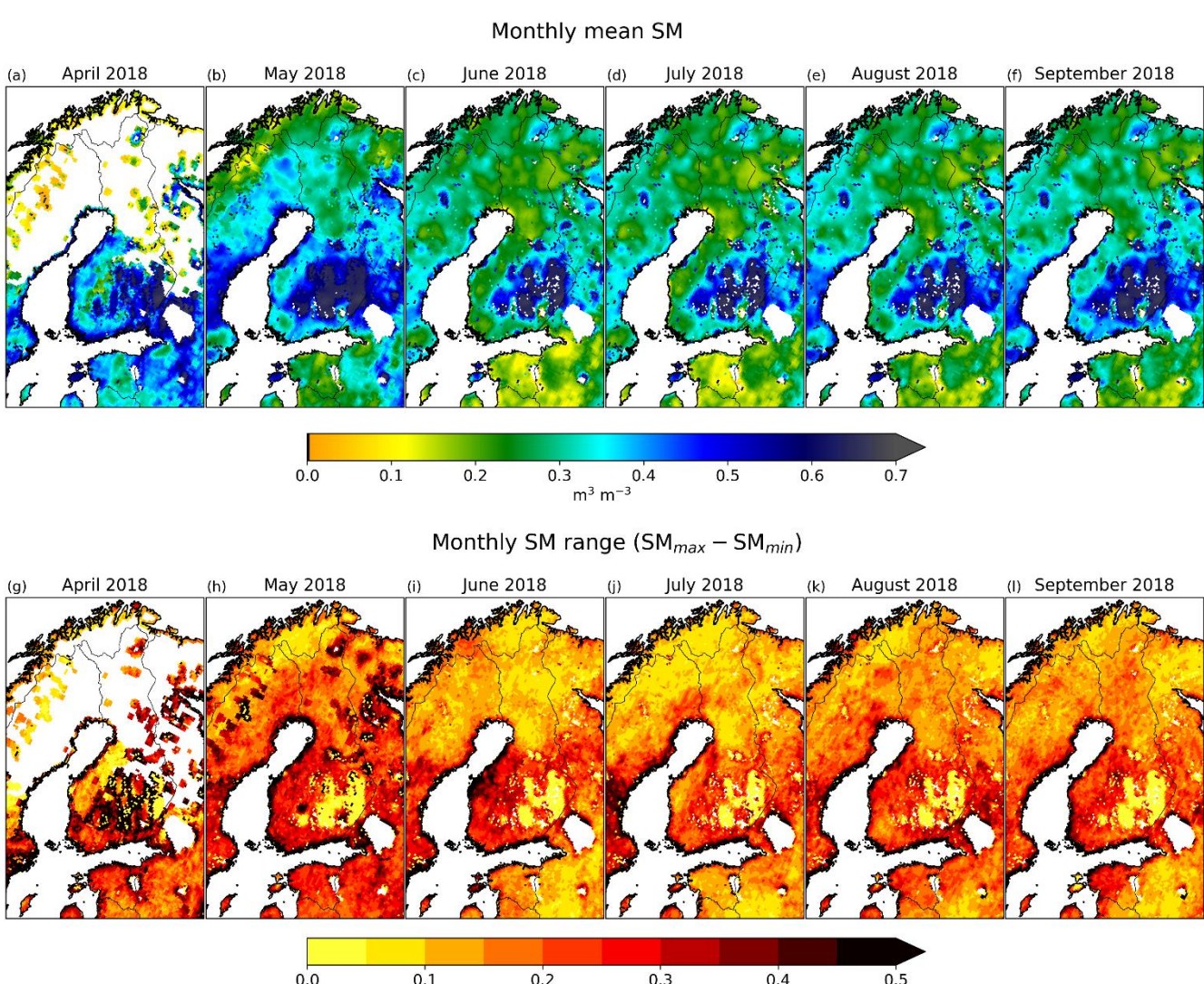

**Figure 4. (a-f) Monthly mean SM and (g-l) monthly SM range in 2018.**

## 3 Results

### 3.1 SM Mean and Variability

Figures 4a-f show the monthly mean SM for April through September in 2018. In April, large areas in Finland remain covered in snow; therefore, no SMAP observations are available. SMAP does not retrieve soil moisture, if the snow areal

fraction within a SMAP grid cell exceeds 50% based on the National Oceanic and Atmospheric Administration (NOAA) Interactive Multisensor Snow and Ice Mapping System (IMS) database (O'Neill et al., 2021a). In the southern regions, snow has already melted, leading to very high SM levels. Snow continues to melt in the northern areas during May, and as a result, SM levels remain high throughout the month. In June, July, and August, after decreasing from the snow-melt levels, the mean SM remains relatively stable but begins to increase again in September. In general, SM is higher in the southern parts

of Finland compared to the north. This agrees well with the spatial distribution of annual P; the soil contains more water in areas that get more precipitation. Additionally, Fig. 4 indicates that SMAP SM retrieval tends to saturate near water bodies, particularly in the southeastern parts of the country.

The range of SM retrievals for each month, i.e. the difference between maximum and minimum SM values, follows a

somewhat similar spatial distribution as the mean values (Figs. 4g-l). The smallest range is in northern Finland and the range increases towards the south. The lake-filled parts of southeastern Finland have a range of 0 because of the saturated SM retrievals. We further analyzed how the observed minimum ($SM_{min}$) and maximum ($SM_{max}$) values within each grid cell depend on soil properties (Fig. S2 in the Supplementary material). $SM_{min}$ shows little variation across different soil textures. In contrast, $SM_{max}$ tends to decrease in sandier soil and increase with higher clay or silt content. However, it is important to

note that the variability in soil texture across the study area is relatively limited (Fig. S1).

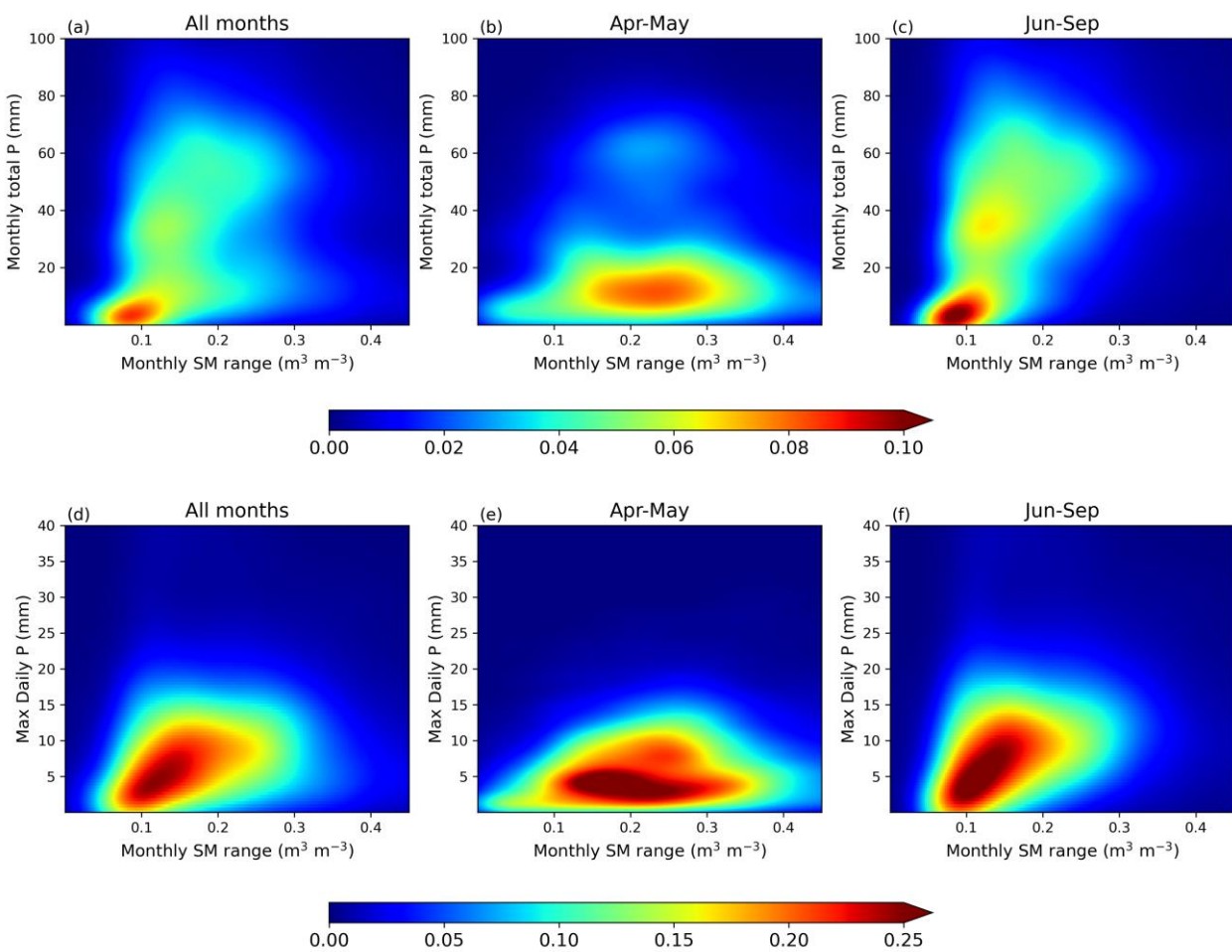

**Figure 5. Density plots for monthly SM range against monthly P (a-c) and maximum daily P (d-f).**

We examined how the monthly SM range in each grid cell correlates with the monthly total P (Figs. 5a-c) and with the maximum daily P throughout the month (Fig. 5d-f). In addition to analyzing these relationships over the entire study period, we also analyzed them separately for April-May, when snowmelt impacts SM, and for June-September. During April and May (Figs. 5b and 5e), we found no correlation between the SM range and either the total monthly P or the maximum daily P. This finding aligns with what was illustrated in Fig. 4: the melting snow tends to raise SM levels, consequently increasing

the SM range. From June onwards (Figs. 5c and 5f), there is a detectable dependency between the variables, with higher daily or monthly P leading to an increase in SM range. Generally, the monthly P values are low, and in most grid cells, they do not exceed 20 mm. However, there are specific grid cells where the monthly P can reach nearly 100 mm, and in these areas, the SM range also exhibits higher values.


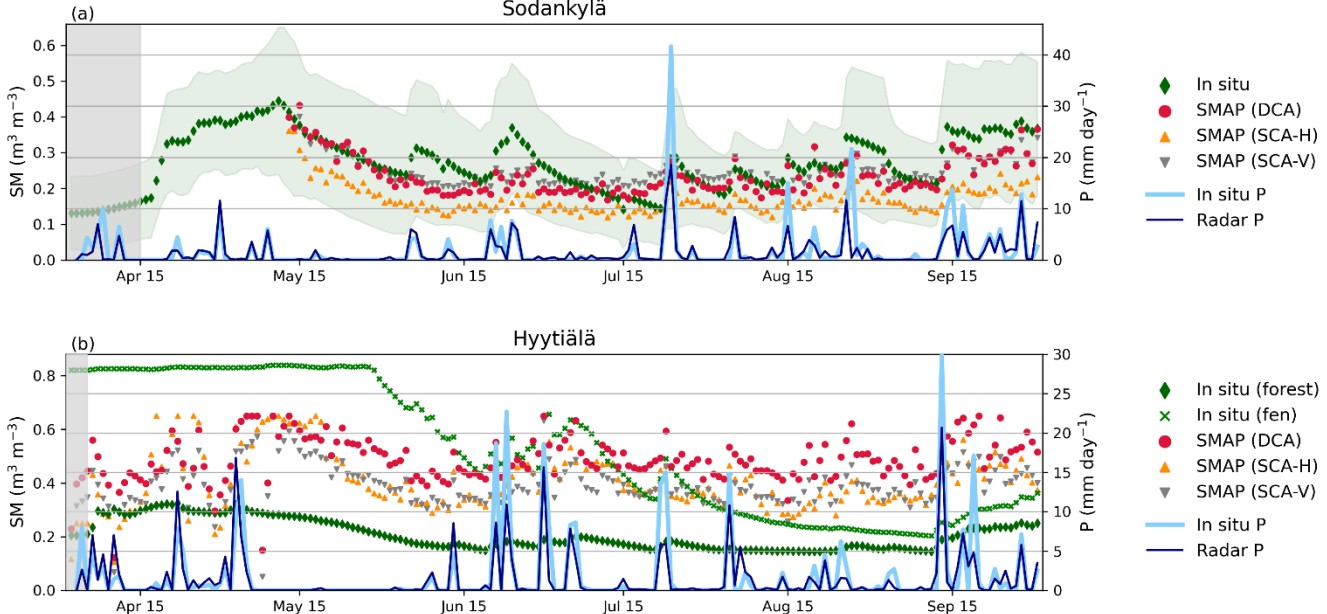

**Figure 6. Timeseries of SM and P in Sodankylä and Hyytiälä in 2018. The green shaded area indicates the standard deviation of the averaged SM measurements in Sodankylä, and the grey shaded area indicates the days when the soil temperature is below 0 °C.**


Figure 6 shows time series data of SMAP SM estimates alongside in situ SM and P measurements and daily radar-based P rates for Sodankylä and Hyytiälä in 2018. The timeseries for Saariselkä, Kenttärova, and Värriö are in the Supplementary material (Fig. S3). In Sodankylä, there is a general agreement between the SMAP estimates and the in situ measurements averaged across the SMAP footprint. In April and early May, only in situ SM measurements are available, as snow covers

over 50% of the grid cell. During the first half of April, the soil remains frozen, which decreases the permittivity measured by the sensor, thus leading to very low SM values. By mid-April, the daily mean air temperature rises above 0 °C (FMI, 2024b) consequently, causing the snow to begin melting. The soil temperature rises above 0 °C and with the combination of melting snow and rainfall, SM values show a clear increase during the last half of April. Snow completely melts by May 13th, which is also when SM reaches its maximum value. During mid to late May, dry conditions prevail, resulting in a

decrease in SM, which both the SMAP and in situ measurements similarly detect. The SCA-H-based SM estimates are slightly lower, while the DCA and SCA-V-based values are more consistent with each other. Later in the study period, in situ SM measurements exhibit slightly greater variability compared to SMAP estimates.

In Hyytiälä, the datasets show larger discrepancies. The Hyytiälä grid cell is flagged with both water and dense vegetation,

complicating satellite-based SM retrieval. There are two in situ SM measurement sites within the grid cell, and they show

notable differences: the site located in the forest consistently detects low values throughout the study period, while the site in the fen initially detects much higher values, which decrease as summer progresses into fall. Initially, the SMAP retrievals fall between the two in situ measurements, but they show higher values by the end of the study period. Based on the in situ measurements, it is evident that the SM can vary notably within a relatively small area (see also, e.g., Famiglietti et al., 1999; 2008). This also considerably impacts satellite-based measurements and complicates SM retrieval but also makes the comparison between satellite-based and in situ SM measurements more challenging and possibly less reliable (e.g., Cosh et al., 2004; Chen et al., 2019). Additionally, the timeseries occasionally show a weak or unclear correlation between SM and P. Minor discrepancies are also observed between radar-based and in situ P measurements, likely due to differences in spatial resolution. In Finland, particularly during summer, precipitation often occurs as localized showers. As a result, rainfall may go unrecorded at the in situ measurement site despite occurring elsewhere within the SMAP grid cell. Conversely, intense rainfall at the in situ site may appear diminished when averaged across the broader SMAP footprint, leading to discrepancies between in situ and radar-based P estimates.

## 3.2 SM2RAIN

To estimate P from SM measurements, we employed the SM2RAIN algorithm, which treats soil as a natural rain gauge. SM2RAIN has been shown to reliably estimate rainfall at temporal resolutions four to six times coarser than that of the input SM data (Brocca et al., 2013; Massari et al., 2014). Given that we used daily SM data as input, we present results for 6-day accumulated precipitation. Figure 7 shows 6-day accumulated P for three selected periods. During the first period (Figs. 7a-d), two regions exhibit high rainfall rates: one in the north and another one in the south. While SM2RAIN generally identifies the correct rainfall locations, estimating precipitation intensity accurately remains more challenging. In the northern region, all the SM2RAIN algorithms (DCA, SCA-V, and SCA-H) tend to underestimate rainfall, whereas in the south, the algorithm more effectively captures the highest intensities. Additionally, all the SM2RAIN algorithms detect rain near the northernmost edge of the study area, which is not visible in the radar data. This discrepancy is likely due to the long distance from the radar; when far away, the radar has difficulty accurately detecting rain, making comparisons less reliable. In eastern and southeastern Finland, the presence of numerous lakes complicates SM retrieval, leading to inaccuracies in rainfall detection. The SMAP single-channel algorithms (Figs. 7c-d) perform better near water bodies, and particularly the SCA-H-based SM2RAIN algorithm is able to detect the southern rainfall area correctly.

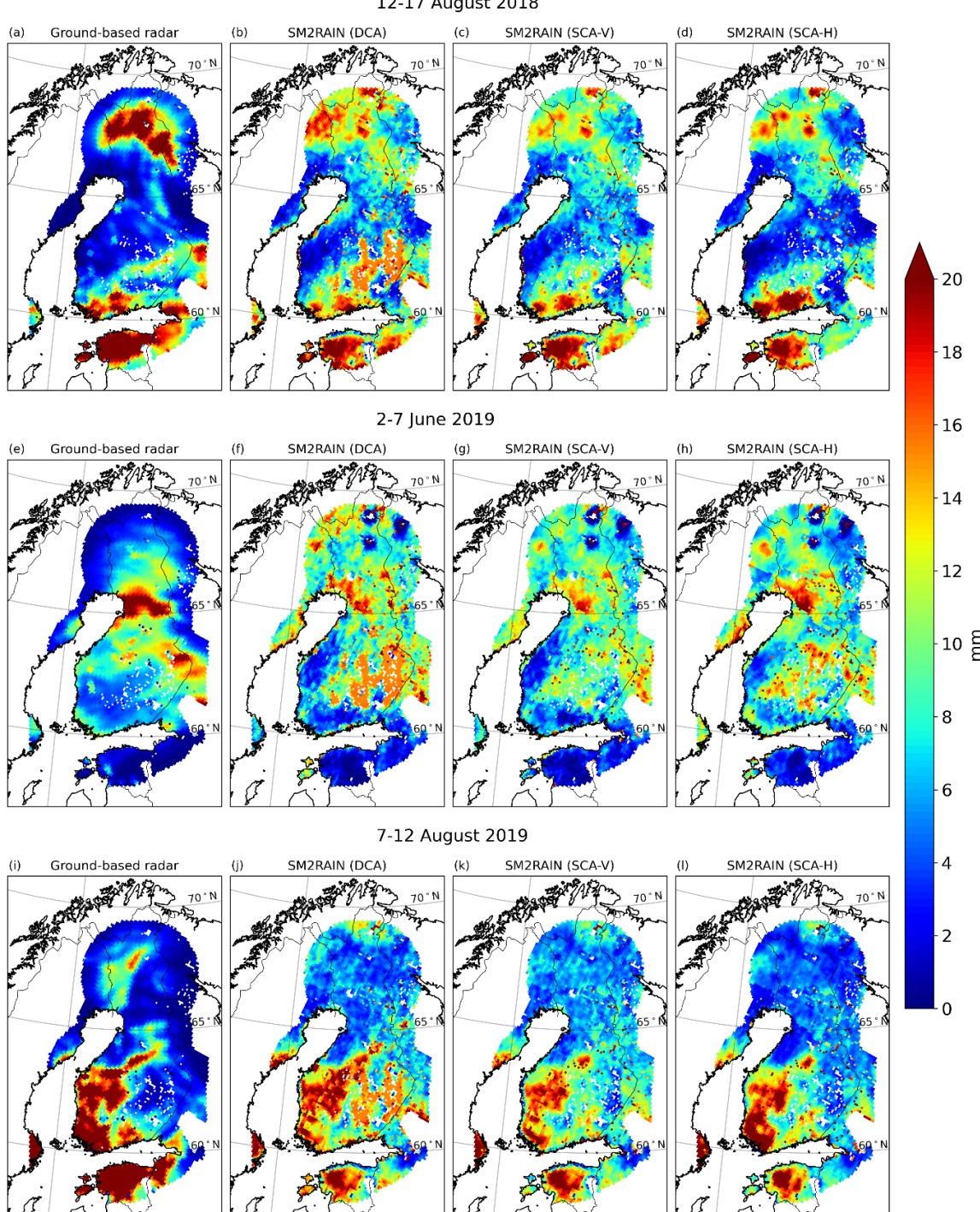

**Figure 7. 6-day accumulated P based on radar data and SM2RAIN algorithm for three selected periods.**

During the second period (Figs. 7e-h), SM2RAIN algorithms indicate the highest intensities around latitude 65 °N, which is

consistent with the radar data. However, the peak intensities are slightly underestimated by the SM2RAIN algorithm. Also, SM2RAIN detects light rainfall across a large area of Finland, overestimating the spatial extent of light rain compared to the radar data. The analysis for the third period (Figs. 7i-l) shows results similar to the previous ones. Overall, while the SM2RAIN algorithms can usually identify areas of rainfall accurately in most cases, estimating the intensity of P remains more challenging. Figure 8 illustrates the bias for 6-day accumulated P (SM2RAIN minus radar-based P) for light, moderate,

and heavy rainfall. For thresholds, we used FMI's definitions for light, moderate and heavy rainfall multiplied by six. For light rain, the median bias is positive (3.8 mm for DCA), indicating an overestimation by SM2RAIN. In contrast, for heavy rainfall, the median bias is markedly negative (-16.5 mm for DCA). These results further highlight that the SM2RAIN algorithm tends to overestimate light rain and underestimate heavy rain, while for moderate rain, the median bias is closer to zero (-0.7 mm for DCA).


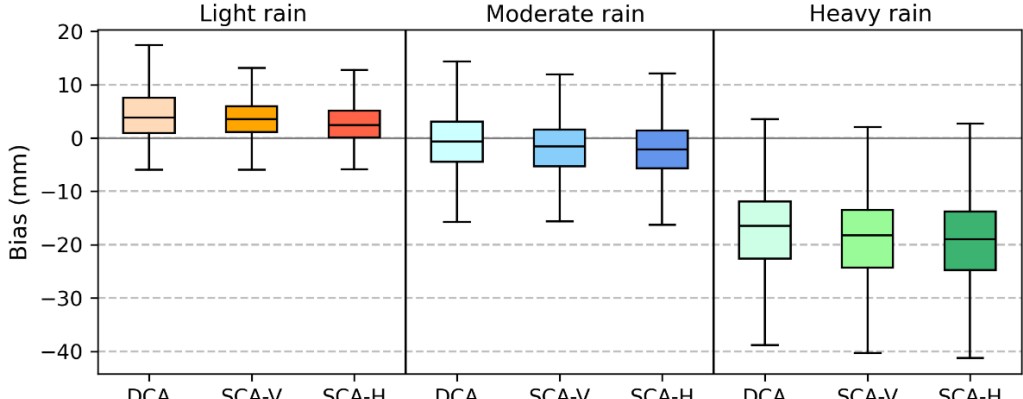

**Figure 8. Box plot of the biases for 6-day accumulated P (SM2RAIN minus radar-based P) for light, moderate, and heavy rainfall for the year 2019.**

## 370 3.3 Drydowns

To identify drydowns, we first analyzed the impact of using different thresholds for daily P. The choice of threshold notably influences both the number of identified drydowns and their maximum length (Fig. S4). Specifically, using a lower threshold results in a decrease in both the number and length of drydowns. When applying the highest threshold (4.5 mm day$^{-1}$), the number of drydowns identified is nearly the same as when no threshold is used (Figs. S4a-b). A stricter threshold leads to a

noticeable decline in the number of drydowns (Figs. S4c-d). In terms of the maximum drydown length (Figs. S4e-h), the patterns are similar: the stricter the threshold, the shorter the maximum drydown length is. This was to be expected, as heavy rain would most likely result in a clear increase in SM which would truncate the drydown regardless of using P information.

For lighter rain, the changes in SM are less pronounced so using a very strict P threshold can truncate drydown when no increase in SM is detected. Since only drydowns lasting at least four days are considered, using a very strict P threshold thus

leads to fewer and shorter drydowns. The spatial distributions for both variables follow a consistent pattern, with the highest number of drydowns and the longest drydowns occurring in the southern regions and progressively becoming lower and shorter towards the north (Fig. S4).

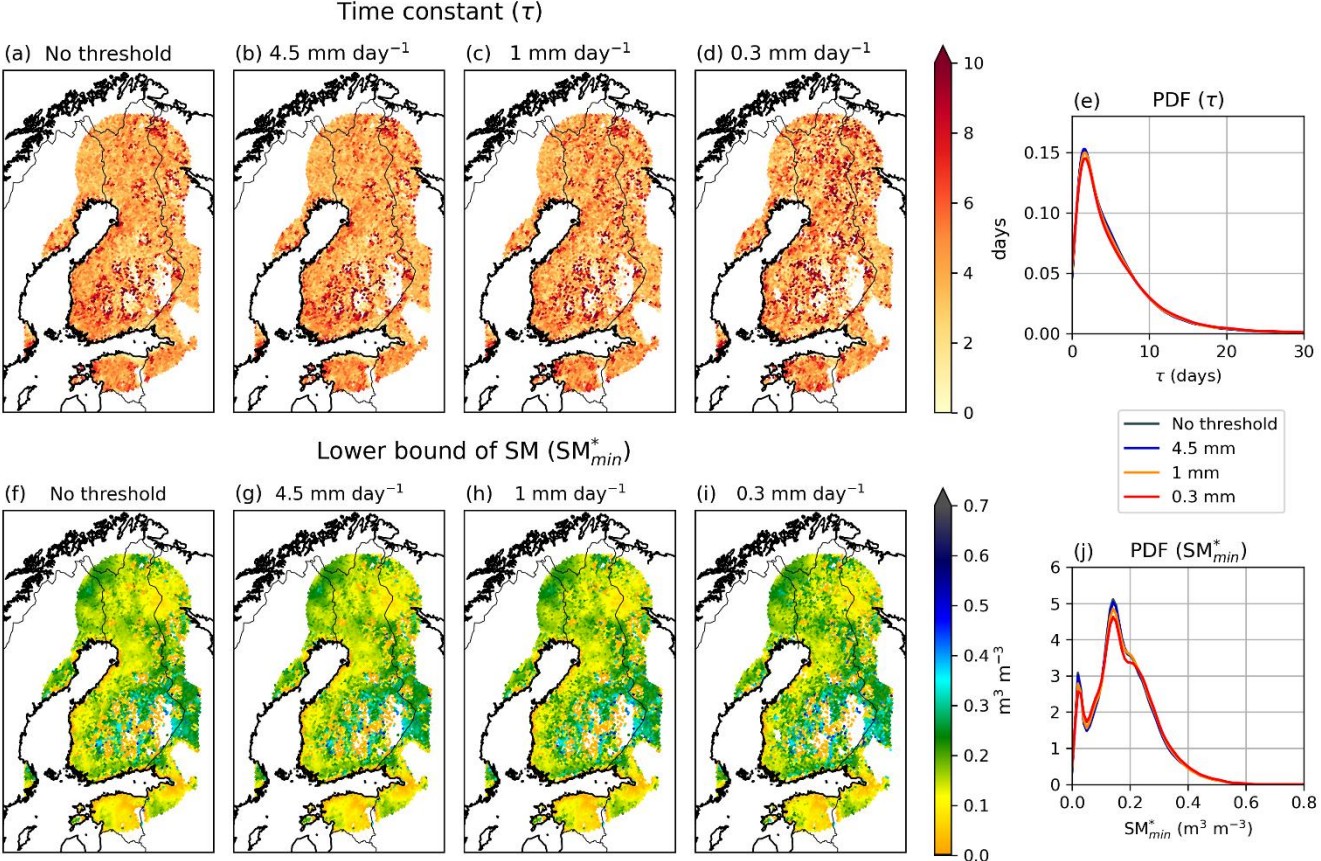

**Figure 9. The median time constant and the median lower bound of SM using different P thresholds when identifying drydowns.**

The time constant $\tau$ remains relatively consistent across different thresholds (Fig. 9). When applying the lowest threshold (0.3 mm day$^{-1}$), the median value is slightly higher in certain areas compared to the higher thresholds; however, the differences between the probability density functions (PDF) are negligible. The lower bound of SM is also unaffected by the

selected P threshold. The values show spatial variability, but the variability is similar regardless of the threshold used. The PDFs for both variables also show similar results for each threshold. The median $\tau$ for all cases is 4.0 days and the median $SM^*_{min}$ is 0.17 m$^3$ m$^{-3}$, respectively. Our analysis indicates that while the chosen threshold influences the number of

identified drydowns and the maximum duration of a drydown, it does not affect τ or the modeled lower bound of SM. Consequently, we have proceeded with the following analysis without applying any P threshold.


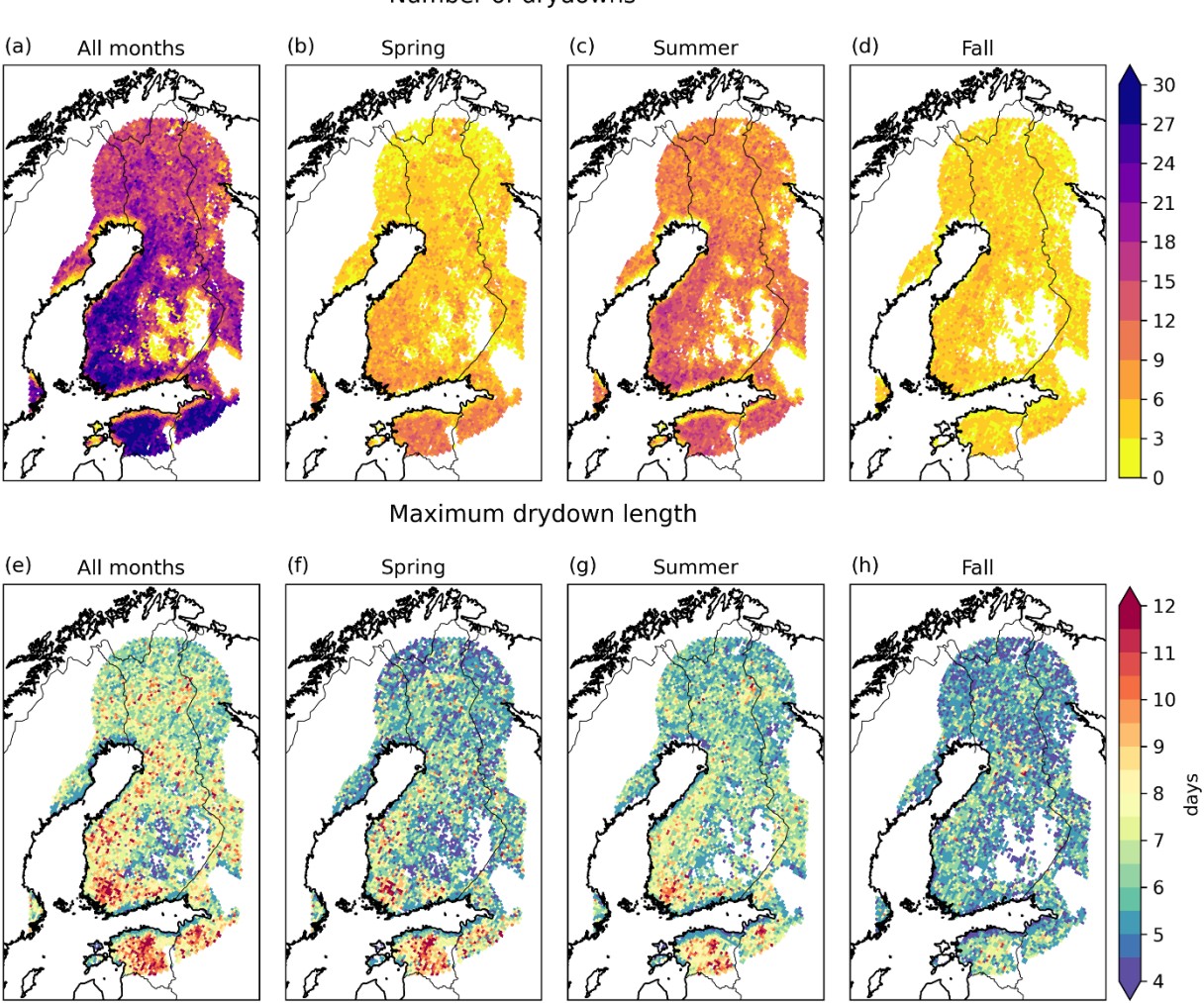

**Figure 10. The number of drydowns and the maximum drydown length for the entire study period and for spring, summer, and fall.**

Figure 10 shows the number of drydowns and the maximum drydown duration in each grid cell for the entire study period, as well as separately for each season: spring (April-May), summer (June-August), and fall (September). The number of identified drydowns shows seasonal variation, which is expected due to the differing lengths of each season. Therefore, the summer season, being the longest, experiences the highest number of drydowns. Overall, the number of drydowns tends to be highest in the southwest region and decreases towards the north. The maximum length of drydowns also varies by season

(Figs. 9f-h), typically becoming shorter as summer progresses into fall. Increased rainfall in the fall likely affects the length of the drydown periods. The spatial distribution of maximum drydown length follows a somewhat similar distribution to the number of drydowns, with the highest values found in the southwest and gradually decreasing to the north. The median drydown length across the entire study area ranges from 5 to 6 days, while the maximum length varies from 12 to 20 days depending on the season, respectively.


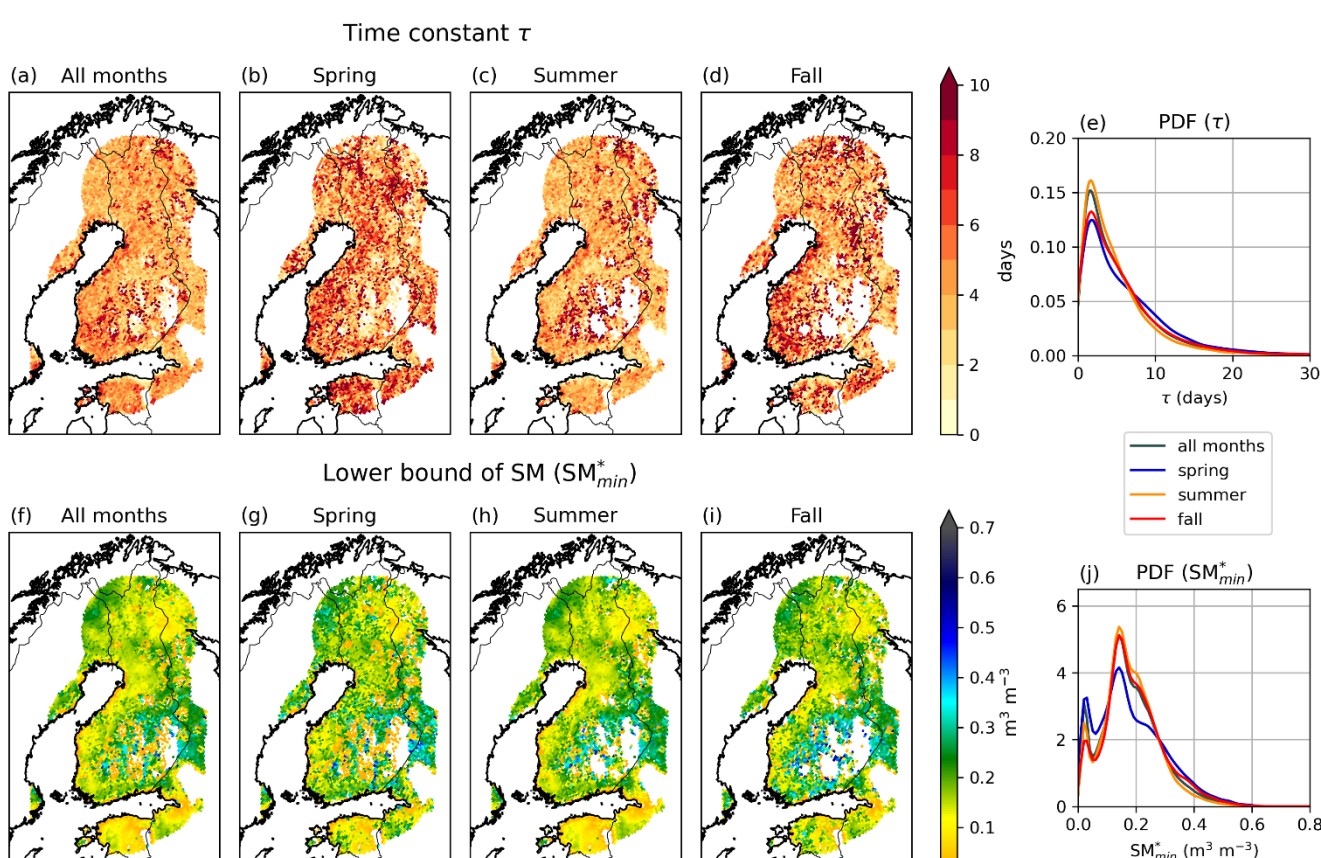

**Figure 11. The median time constant and the median lower bound of SM for the entire study period and for spring, summer, and fall.**

Figure 11 shows the spatial distribution of the median of $\tau$ and the modeled lower bound of SM, and PDF for both variables throughout the entire study period and separately for the spring, summer, and fall seasons. $\tau$ in each grid cell varies mostly between 1 to 10 days, with slightly higher values in spring and fall. The spatial distribution does not reveal any clear patterns in any of the seasons. The distribution (Fig. 11e) is highly positively skewed with a heavy tail. Across all seasons, the mode is 1.6 days, while the median ranges from 3.6 to 4.5 days, depending on the season. Unlike $\tau$, the lower bound of SM exhibits

some spatial variability: the values tend to increase slightly near water bodies (Figs. 11f-i). However, near the water bodies,

SM*$_{min}$ can occasionally present very low values, particularly in spring. The PDF (Fig. 11j) demonstrates that the distribution of SM*$_{min}$ displays a somewhat similar pattern across all seasons, with minor variations. The distribution of SM*$_{min}$ is positively skewed, with a mode of 0.14 m$^3$ m$^{-3}$ and a median ranging from 0.16 m$^3$ m$^{-3}$ to 0.18 m$^3$ m$^{-3}$. Furthermore, SM*$_{min}$ exhibits an additional lower peak at 0.02 m$^3$ m$^{-3}$, which is most pronounced in spring and decreases during summer and fall. The SMAP SM retrievals have a lower limit of 0.02 m$^3$ m$^{-3}$ (O'Neill et al., 2021a), which may cause an artifact in the results.

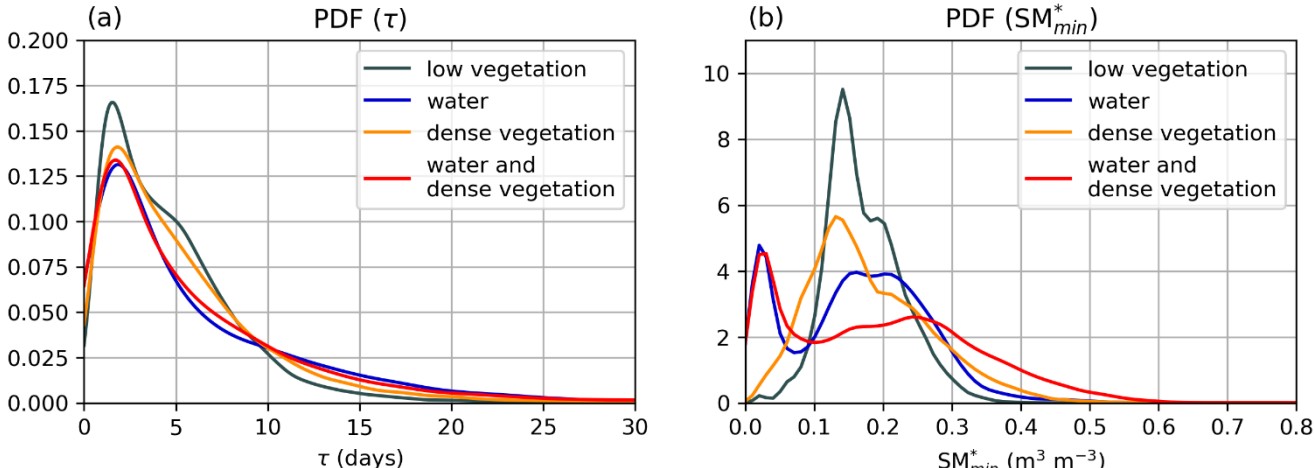

**Figure 12. The probability density functions for $\tau$ and SM*$_{min}$ in each class defined in Fig. 2c.**

Figure 11 indicates that SM*$_{min}$ is particularly influenced by surface classification, as its spatial variability closely resembles those shown in the surface flag map (Fig. 2c). Therefore, we analyzed the distribution of both $\tau$ and SM*$_{min}$ separately for each surface class defined in Fig. 2c. As expected, $\tau$ does not vary considerably across the classes (Fig. 12a). The peak is slightly higher for the low vegetation class; however, overall, $\tau$ displays a similar distribution in all classes. In contrast, SM*$_{min}$, varies notably across the classes (Fig. 12b). The low vegetation class exhibits the highest peak, with no detectable lower peak. The dense vegetation class also exhibits a single peak, which is slightly lower than that of the low vegetation class. The other two classes, "water" and "water and dense vegetation", show a mode at 0.02 m$^3$ m$^{-3}$, along with a lower and less distinct peak at higher values. Since 0.02 m$^3$ m$^{-3}$ is the lower limit of SMAP SM retrievals, it may introduce an artifact in the results. Figure 12b indicates that water bodies particularly complicate the SM retrieval. While dense vegetation alone does not pose as much of a challenge, its combination with water makes SM retrieval more difficult.

In addition to surface classification, we examined the influence of soil properties on the parameters SM*$_{min}$ and $\tau$. Specifically, we investigated whether sand, silt, and clay fractions, as well as bulk density and soil porosity, have any effect

on these variables, and whether seasonal variability plays a role. The analysis showed no clear impact of soil properties on

$SM*_{min}$ or $\tau$, and no significant seasonal variation was observed (Figs. S5-S6 in the Supplementary material). However, it is important to note that soil texture within the study area exhibits limited variability (Fig. S1). A broader study area with greater heterogeneity in soil characteristics may reveal more pronounced effects. This possibility could be explored in future research.

## 4 Discussion

We have analyzed the relationship between SM and P and investigated whether satellite-based SM data can be used to estimate P rates by employing the SM2RAIN algorithm. Additionally, we examined the patterns of SM drydowns across Finland.

The results indicate a generally strong correlation between SM and P, with SM levels rising during rainfall events and
decreasing during dry periods. However, during April and May, snowmelt increases the SM levels across the study area (Figs. 4a-b), resulting in a less clear relationship between SM and P (Fig. 5b and 5e). Despite low overall rainfall rates in spring, the monthly range of SM remains relatively high. From June onward, a stronger relationship between SM and P emerges (Figs. 5c and 5f).

Using the SM2RAIN algorithm, we evaluated the potential of estimating P from satellite-based SM data. Previous studies indicate that SMAP data typically perform well as input for the SM2RAIN algorithm (Koster et al., 2016; Miao et al., 2023a), although forested areas complicate retrievals (Lai et al., 2022). Our analysis shows that while the algorithm accurately identifies the spatial extent of P, it struggles with intensity detection (Fig. 7). Specifically, the SM2RAIN algorithm tends to overestimate light rain and underestimate heavy rain (Fig. 8), aligning with findings from prior research
(Koster et al., 2016; Tarpanelli et al., 2017). Additionally, as Figs. 5b and 5e showed, the relationship between P and SM is more complex during snowmelt season. In spring, soil wetting due to snowmelt can complicate the algorithm's accuracy, as it can misinterpret increased SM from snowmelt as rainfall, leading to inaccuracies in P estimates. Therefore, we restricted our analysis to the period from June onwards to exclude the month affected by snowmelt. Nonetheless, future research could more thoroughly examine the dynamics between P and SM during the melt season to analyze whether it is possible to
estimate P from SM during melt season. This approach could also benefit from using additional satellite-based information on melt onset and timing (e.g. Anttila et al., 2018; Wang et al., 2013).

In this study, we have used a simple form of the SM2RAIN algorithm, which includes only SM variations and the drainage term. Previous research has demonstrated that this simplified formulation is generally sufficient, as neglecting

evapotranspiration and surface runoff terms does not weaken the algorithm's performance (Brocca et al., 2015). However, future studies may benefit from including these additional terms to potentially enhance the accuracy of P estimates.

The coarse spatial and temporal resolution of SMAP presents challenges in detecting small-scale phenomena. The original resolution of the radar data is considerably finer, which can lead to inaccuracies in comparisons, even after averaging the radar data to match the SMAP resolution. The spatial resolution of SMAP does not capture the systematic small-scale SM patterns. Figure 6b showed that SM can have systematically very different SM conditions within a single grid cell, making it challenging to relate the area-averaged SM measured by SMAP to the local SM conditions. Furthermore, the temporal resolution, ranging from one to two days, complicates the detection of rapidly moving rainfall areas. As Fig. 1 illustrated, the satellite-based SM data cannot capture all the variability in SM. Therefore, detecting small-scale precipitation events, which are common in Finland during summer, is challenging with the SMAP data. Additionally, the presence of water bodies introduces artifacts in SM retrieval, affecting the accuracy of P estimates. Despite these challenges, the results are promising, and our findings highlight the potential of satellite-based SM data for estimating P rates. This has considerable implications for improving rainfall estimation in regions with sparse ground-based observations, enhancing weather forecasting, and contributing to climate models.

We analyzed SM drydown patterns by fitting an exponential model to each SM drydown period, from which we estimated the exponential decay time scale ($\tau$) and the lower bound of SM ($SM^*_{min}$). We tested three specific P thresholds and studied whether they had an impact on the drydowns. The results showed that the threshold affected both the number of drydowns and the maximum length of a drydown. However, they did not influence $\tau$ or $SM^*_{min}$, which is why we conducted most of the analysis without using any P threshold.

Our findings indicate that $\tau$ exhibits minimal spatial or temporal variability (Fig. 11). The distribution of $\tau$ is positively skewed, with a mode of 1.6 days and a median of 4.0 days. The median agrees well with in situ measurements, which report a median value of 4-6 days (Shellito et al., 2016; Raoult et al., 2021). In contrast, $SM^*_{min}$ shows more notable spatial variability. When we analyzed the distribution across different surface classes (as defined in Fig. 2c), we observed distinct differences (Fig. 12). Grid cells containing water bodies (classified as "water" and "water and dense vegetation") show a mode in $SM^*_{min}$ at 0.02 $m^3$ $m^{-3}$, alongside a lower and less distinct peak at higher values. Since 0.02 $m^3$ $m^{-3}$ is the lower limit for SMAP SM retrievals, this may introduce an artifact in the results. Conversely, grid cells without water bodies ("low vegetation" and "dense vegetation") display more consistent distributions, each with a single peak around 0.13 $m^3$ $m^{-3}$. These differences in distribution suggest that surface conditions significantly influence the SMAP SM retrievals. Specifically, this result indicates that water bodies complicate SM retrieval, while dense vegetation alone does not pose as much of a challenge. This conclusion is further supported by previous findings demonstrating SMAP's capability to detect SM beneath vegetation cover (Ayres et al., 2021; Colliander et al., 2020).

In this study, we have used the exponential decay model to define $SM^*_{min}$ and $\tau$, following the methodology from McColl et al. (2017b). While this approach has been widely used (e.g. McColl et al., 2017b; Rondinelli et al., 2015; Shellito et al., 2016), future research could explore alternative decay models. Considering alternative functions could potentially lead to even more accurate characterization of SM dynamics during drydown events.

The datasets used in the analysis generally perform well, but they may cause minor inaccuracies in the analysis. The radar-based P data used as reference are overall accurate (Saltikoff et al., 2010) and agree well with in situ measurements. However, radar accuracy diminishes toward the edges of the study area due to beam curvature (Saltikoff et al., 2010), leading to slight uncertainties in these regions. These uncertainties are minimal, as the study area was defined based on radar distance (Fig. 2a). SMAP has also been found to perform generally well, often exceeding the accuracy of SM estimates based on SMOS and ASCAT (Colliander et al., 2023; Chen et al., 2018). However, some differences exist between the satellites, and further studies using different satellite datasets could assess and potentially refine our findings.

Based on the analysis, water bodies complicate satellite-based SM estimates. In northern Finland, SM is measured in specific areas during April (Fig. 4a), even though much of the region remains covered in snow at that time. These areas correspond closely with areas identified as water bodies (Fig. 2c), indicating that the water fraction within those grid cells ranges from 5% to 50%. Furthermore, these areas show a noticeable variation in monthly SM in April and May (Figs. 4g-h). In spring, many lakes in Finland remain frozen which likely influences the anomalous SM measurements in these regions. In summer, SMAP SM retrievals in turn tend to saturate near water bodies, particularly in southeastern parts of Finland (Fig. 4). The analysis also shows that the single-channel algorithm performs better near water bodies. Recent studies on SMAP performance in forests show that SMAP is able to detect SM variability under the canopy (Colliander et al., 2020b). This analysis is consistent with our findings: the analysis shows that water bodies complicate SM retrieval, while dense forests do not pose a similar challenge. This finding also suggests that the methods used in this study could be applied across the Arctic, as water bodies cover only about 6% of the Arctic land surface (Paltan et al., 2015). However, another factor that may complicate SM retrievals in the Arctic, and consequently affect the performance of the SM2RAIN algorithm, is the extensive permafrost coverage across the region (Aalto et al., 2018). Thawing permafrost can introduce notable uncertainties in P estimates derived from SM2RAIN (Miao et al., 2023b). This interaction between permafrost dynamics and SM2RAIN performance represents a valuable topic for future research, particularly in the context of a warming Arctic.

Although this study primarily examined how P affects SM, it is important to note that the relationship can also work in the opposite direction. Specifically, SM influences heat fluxes from the ground thus impacting the atmospheric humidity and temperature profiles, which in turn can affect P rates (Seneviratne et al., 2010; Tuttle and Salvucci, 2016). While this was not the focus of the current study, it could be an area of interest for future research.

**5 Conclusions**

We have analyzed the relationship between SM and P, assessed the feasibility of estimating P from SM, and examined SM
drydown patterns across Finland from April to September 2018-2019. This study was based on SM data from NASA's
SMAP satellite and used ground-based weather radar data and in situ SM and P measurements as reference data. The main
findings of this study are as follows:

- There is generally a good agreement between SM and P, with SM levels rising during rainfall events and decreasing
  during dry periods. However, in April and May, the correlation is weaker due to snowmelt, which increases SM
  levels. From June onward, a stronger relationship between SM and P emerges.

- Using the SM2RAIN algorithm, we evaluated the potential of estimating P from satellite-based SM data. Overall,
  the analysis indicates that the algorithm detects the area of rainfall accurately, but estimating the intensity of P
  remains more challenging. Specifically, the algorithm tends to overestimate light rain (median bias 3.8 mm for 6-
  day accumulated P) and underestimate heavy rain (median bias -16.5 mm for 6-day accumulated P), which is
  consistent with prior research.

- We analyzed SM drydown patterns by fitting an exponential model to each SM drydown period, from which we
  estimated the exponential decay time scale ($\tau$) and the lower bound of SM ($SM^*_{min}$). Our findings indicate that $\tau$
  exhibits minimal spatial or temporal variability. The distribution of $\tau$ is positively skewed, with a mode of 1.6 days
  and a median of 4.0 days, consistent with other studies. The distribution of $SM^*_{min}$ is also positively skewed, with a
  mode of 0.14 $m^3\ m^{-3}$ and a median of 0.17 $m^3\ m^{-3}$.

- The analysis also shows that surface conditions significantly influence the SMAP SM retrievals. Specifically, our
  findings indicate that water bodies complicate SM retrieval, while dense vegetation alone does not pose as much of
  a challenge. This result suggests that the method used in this study could be applied across the Arctic because only a
  small fraction of the Arctic land surface is covered by water bodies.

**Data availability**

The SMAP data are available at https://nsidc.org/data/spl3smp_e/versions/6. The radar data and in situ P measurements are
available at https://en.ilmatieteenlaitos.fi/open-data. The in situ SM measurements are available at
https://ismn.earth/en/dataviewer/ (Sodankylä and Saariselkä), https://doi.org/10.5281/zenodo.10820563 (Kenttärova), and
https://smear.avaa.csc.fi/ (Hyytiälä and Värriö).

**Author contributions**

KK performed the analysis and produced the figures with substantial contributions from AC. KK wrote the original draft. Both authors contributed to manuscript review and editing.

**Competing interests**

The authors declare that they have no conflict of interest.

**Acknowledgments**

This work was financially supported by the Research Council of Finland in the project SNOCAP (341845). A contribution to this work was made at the Jet Propulsion Laboratory, California Institute of Technology, under a contract with the National Aeronautics and Space Administration.

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
