# Peer review of "Figure S1. Timeseries of SM and radar-based P measurements in Saariselkä, Kenttärova, and Värriö."

_EGUsphere, 2025_

## Author Comment (AC1)

The review comments are shown in black, and the author responses are in red.

We would like to thank the reviewer for reviewing our manuscript. Please find below our responses to the comments.

The paper reads well and is interesting. It covers a topic not always fully assessed but I never the less have questions about a few facts. For this reason I think maybe the manuscript should be resubmitted (hence my reject suggestion) to account for the issues raised in the first two points. But I agree it could be only major revisions should the authors be able to address the 2 issues.

Maybe the most striking is why is it important to assess rainfall in Arctic environment where – to my understanding – most of the water comes under the form of solid precipitation. In other words what is the real impact of assessing exactly liquid precip and how the errors in liquid precip relate to uncertainties in solid precip.

Precipitation in both solid and liquid form is essential for the Arctic region. Depending on the specific location, over half of the total annual precipitation can be in liquid form. Figure 1 illustrates the proportion of rainfall relative to total precipitation based on ERA5 data from 2000 to 2024, both for the entire year and specifically for the months of April to September, which are the focus of this study. The figure shows that in extensive areas of the Arctic, liquid precipitation constitutes over half of the annual total precipitation. For Apr-Sep, the fraction further increases.

Additionally, research indicates that total precipitation in the Arctic has increased by 9% during 1971-2019, which is driven by a 25% increase in rainfall (AMAP, 2021). AMAP (2021) also states that while precipitation (both solid and liquid) estimates are available from the reanalyses, their quality for the Arctic region remains poorly established. Therefore, there is a clear need to improve the estimates of liquid precipitation in the Arctic. We will elaborate on this more in the manuscript.

[Figure]

**Figure 1. The proportion of rainfall relative to total precipitation based on ERA5 data from 2000 to 2024.**

A second big question is about the neglecting of run off. (line 169). To me, but I might be wrong especially for Arctic areas, most of the run off occurs during rainfall or immediately after, so it cannot be negligible.

First, we would like to clarify that the neglected runoff term only includes surface runoff, whereas subsurface runoff is included in the SM2RAIN algorithm. Specifically, the drainage term (Eq. 2 in the manuscript), contains the subsurface runoff. We will clarify this in the manuscript.

Brocca et al. (2015) studied the impact of various terms on the precipitation (P) estimates derived from the SM2RAIN algorithm. Their research demonstrated that soil moisture variations and the drainage term are the most significant contributors, accounting for over 90% of the simulated P estimates. The study concluded that neglecting evapotranspiration and surface runoff terms does not weaken the algorithm's performance. Additionally, the use of satellite data with relatively coarse resolution further reduces the impact of surface runoff. The portion that does not infiltrate—due to factors such as impervious land cover or soil—may re-infiltrate downstream within the scale of a SMAP grid cell (Brocca et al., 2019).

Nevertheless, we acknowledge that limitations exist in the SM2RAIN algorithm. In future studies, analyzing more complex versions of the SM2RAIN algorithm would be worthwhile, although that is beyond the scope of this study. As this is the first study to analyze SM2RAIN's performance in the Arctic and in regions with seasonally frozen ground, we consider that the current simple form of the SM2RAIN algorithm is adequate. Now that the performance has been established in this study, the next step would be to study more complex versions of SM2RAIN and its performance in future studies. We will add discussion about the limitations and future perspectives in the manuscript.

I also have question on some aspects of the range of soil moisture. Maps show SM in excess of 0.7 m3/m3 but (figure 4) but I doubt the field capacity is higher than 0.5 m3/m3. So even the range seems excessive. It could corresponds to flooded areas but then the range being minimal, it would correspond to water bodies. But surely SMAP sees and flags water bodies (otherwise SM estimates are bound to be wrong) so what is it exactly? The authors might want to elaborate on this as it is most intriguing.

It is correct that SMAP sees and flags water bodies. As mentioned in the manuscript (lines 147-148), a grid cell is marked as water if the water fraction exceeds 5%. If the water fraction exceeds 50%, the SM retrieval is skipped. This means that if the water fraction within a grid cell is between 5% and 50%, a correction is applied and the grid cell is flagged for the high water fraction, but SM is still retrieved. The water bodies affect the quality of SM retrievals and therefore, the grid cells flagged for high water fraction do not have recommended quality (Fig. 2b).

Finland is characterized by an abundance of lakes, as shown in Fig. 2c, where large areas are flagged for high water fraction (light and dark blue areas). However, Finnish lakes are often small and vary in shape and size. Consequently, even though many grid cells are flagged for high water fraction, SM is still retrieved in most of them because the water fraction does not exceed 50%. The number of lakes is especially high in southeastern Finland, which also shows the highest SM values (Fig. 4). The presence of numerous lakes leads to saturated SMAP SM retrievals in these regions.

We acknowledge that water bodies complicate SM retrievals. As mentioned in the manuscript (lines 143-146), Finland is a challenging region for satellite-based SM retrieval, as large areas of the study region are covered with either water, dense vegetation, or both. However, despite the quality concerns, we chose to include all grid cells in our analysis, regardless of the surface flags, to evaluate how well the methods used in this study perform under suboptimal SM retrieval conditions. As the surface conditions affect the SM retrieval, we have identified these regions based on the SMAP surface and quality retrieval flags and divided the study area into four classes (Fig. 2c). We will provide further elaboration on this in the manuscript.

I am not sure also I understand the lower limit of 0.02 m3/m3 for SMAP Does it mean that SMAP SM estimates are never lower than this value? And if yes what is the rationale for this.

The lower limit for SMAP soil moisture retrieval is 0.02 $m^3m^{-3}$, meaning that the soil moisture values are never lower than that value. This limit is established by the SMAP team (O'Neill et al., 2021) and is beyond our control.

I have also a question for the authors on the choice of SM2RAIN. As here are some limitations linked to the assumptions of the approach (such as the one made above). So why use such algorithm? Why not use more robust approaches assimilating SMAP data in a simplistic model to infer rainfall? Of course such approaches require a initialisation through a first guess precip usually from systems such as IMERG. Did the authors consider such approaches which should be more reliable and why did they or did not? I am thinking of Pellarin et al for instance.

While it is true that the algorithm has limitations, one of the main advantages of the SM2RAIN algorithm is that it requires only soil moisture information to estimate precipitation. This is particularly beneficial for our purposes. Our aim is to apply this method across the entire Arctic, and the ability to rely solely on SM information provides a significant opportunity to estimate precipitation in remote Arctic regions. Despite its limitations, the algorithm has shown promising results in other regions. This study aims to investigate whether this approach can be effectively used in the Arctic. We will revise the manuscript to clarify why this approach is being utilized.

Line 112 a point of detail, the native spatial resolution of SMAP is not 36 km (it is an ellipse) but rather 39x47 km according to the SMAP handbook. So this statement is somewhat misleading. As very rightly indicated the 9 km grid corresponds to oversampling but this is not clear in table 1 where the resolution is indicated as being 9 km. The latter should be corrected.

The user guide for the SMAP product states that "the native spatial resolution of the radiometer footprint is approximately 36 km" (Section 1.3.2 in O'Neill et al., 2021). After consideration, we have decided to keep the text on line 112 unchanged to maintain consistency with the product user guide.

The resolution mentioned in Table 1 (9 km) corresponds to the grid resolution of the product used in this study. This table is designed to clearly present the data products used in this study, making it easier for readers to refer to the exact products used in the research. Since the SMAP product used in this study has a grid resolution of 9 km (with radiometer footprint defined native resolution), we believe it is clearer for readers to specify that resolution in the table.

References:

AMAP, 2021. AMAP Arctic Climate Change Update 2021: Key Trends and Impacts. Arctic Monitoring and Assessment Programme (AMAP), Tromsø, Norway. viii+148pp

Brocca, L., Massari, C., Ciabatta, L., Moramarco, T., Penna, D., Zuecco, G., Pianezzola, L., Borga, M., Matgen, P. and Martínez-Fernández, J.: Rainfall estimation from in situ soil moisture observations at several sites in Europe: an evaluation of the SM2RAIN algorithm. Journal of Hydrology and Hydromechanics, (3), 201-209, https://doi.org/10.1515/johh-2015-0016, 2015.

Brocca, L., Filippucci, P., Hahn, S., Ciabatta, L., Massari, C., Camici, S., Schüller, L., Bojkov, B., and Wagner, W.: SM2RAIN‐ASCAT (2007‐2018): global daily satellite rainfall data from ASCAT soil moisture observations, Earth Syst. Sci. Data, 11, 1583‐1601, https://doi.org/10.5194/essd-11-1583-2019, 2019.

O'Neill, P. E., S. Chan, E. G. Njoku, T. Jackson, R. Bindlish, J. Chaubell, and A. Colliander.: SMAP Enhanced L3 Radiometer Global and Polar Grid Daily 9 km EASE-Grid Soil Moisture, Version 6. Boulder, Colorado USA. NASA National Snow and Ice Data Center Distributed Active Archive Center. https://doi.org/10.5067/M20OXIZHY3RJ, 2021.

---

## Author Comment (AC2)

The review comments are shown in black, and the author responses are in red.

We would like to thank the reviewer for reviewing our manuscript. Please find below our responses to the comments.

In this paper, SMAP SM data are analyzed in the Arctic region (specifically in Finland) to assess their capability in estimating SM in this complex environment, explore the relationship between P and SM, estimate precipitation from SM using SM2RAIN algorithm and examine SM drydown pattern from April to September in the years 2018-2019. The analysys shown promising results, with good correlation between SM and P, excluding frozen and snowmelt period; limitations of SM quality in the proximity of water bodies and good spatial accuracy of the SM2RAIN rainfall product (but low intensity accuracy, with overestimation of low rainfall and underestimation of the heavy rainfall). Moreover, the length and the exponential decay time scale of the SM drydown were analyzed, showing stable results both in time and space.

The paper is well organized and well written, the language is fluent and clear.

However I have a few important points that need to be answered before the paper is accepted for publication. Specifically

Major comments:

Lines 179-172 "Using…. 2014)": Not clear. Hence the SM2RAIN algorithm was not calibrated? This approach is not correct: SM2RAIN parameters need to be calibrated to each pixels, as they are dependent from the soil characteristic of the pixels. All the works of Brocca et al. after 2013 relied on this, to overcome the limits of 2013 paper. SM2RAIN should be calibrated against observations (without considering SM data obtained in frozen conditions). If the Authors prefer to not calibrate the data, they should still use some regionalization procedure to obtain SM2RAIN weight for each pixel (e.g. Filippucci et al 2021; 2022, here applied to ASCAT or S1 data). The parameters can change greatly according to the area. Moreover, this approach do not consider SM2RAIN temporal filter T. This would be important to assess the real performance of SM2RAIN in the area.

We will revise the approach and include both calibration and filtering as suggested.

Moreover, SM2RAIN use interpolated daily data, normalized between 0 and 1. Did the Authors applied interpolation and normalization? It should be written and, if not applies, please explain why.

Yes, we have applied both interpolation and normalization. In case of missing SMAP data, the data are linearly interpolated, and a maximum data gap of three days is considered for the interpolation. We will clarify this in the text.

Regarding SM2RAIn quality, the error introduced by the snowmelt was already analyzed in Brocca et al., 2019, which applied mask to sm data during melting (0<T<3) in order to avoid this kind of noises. This should be implemented also here, to assess the very capabilities of NASA SMAP data.

We will implement this method and exclude the melt season from the SM2RAIN analysis.

Line 419 "while dense….challenge": I don't think that this conclusion is supported by this analysis. Dense vegetation show a realistic pattern, but it is not verified that the SM beneath the vegetation is actually sensed. One way to obtain a confirmation would be to use SM2RAIN to evaluate precipitation time series for all the pixels and then assess its performance per pixel (with indices as RMSE or Pearson correlation or BIAS). If the results in the densely vegetated area are similar to those around, then this indicates that the capacity of SMAP to estimate SM under vegetation are good. But I expect to find worse correlation. In general, I understand that

the lack of SM data allow to validate SM just in few areas, but precipitation data are available for all the study areas, hence I encourage to use SM2RAIn to gain more information regarding the SM data quality

It is true that the lack of reference SM data complicates the validation of satellite-based SM retrievals. However, research has shown that SMAP is able to detect soil moisture beneath the vegetation (Colliander et al., 2020; Ayres et al., 2021). Using P data together with the SM2RAIN algorithm to evaluate SMAP SM retrievals would be a worthwhile study, but we consider it being out of scope of this study. Nevertheless, we acknowledge that uncertainties exist and will discuss the limitations in the manuscript.

Minor comments:

Lines 55-57 "While these….2019)": This is not completely true, as, for example, brocca et al. do not exclude artic region, just periods in which the soil is frozen. Check and correct

We will check and edit the text accordingly.

Lines 267-269 "This….site": Indeed, the discrepancies in SM between the two sites could be related to different precipitation pattern in the large SMAP pixel. However, it could be also related to error in SM measurement. Since Radar data have 250 meters spatial resolution, you could compare the radar Precipitation obtained in the pixels nearest to the Hyytiälä stations, to ensure that they are correlated with it and therefore there is no error in the observed SM measurements.

We will make the comparison as suggested.

Line 363 "Figure…. on SM": Not clear, please expand the reasoning behind the sentence

There is a typo in the sentence, and it should read Figure 11 instead of Figure 10. We will correct the typo and also ensure that the sentence is clear for the reader.

Lines 386-387 "Our analysis….Fig. 7),": To be checked after proper SM2RAIN calibration (see major comment #1)

We will check and edit the text accordingly.

References:

Ayres, E., Colliander, A., Cosh, M. H., Roberti, J. A., Simkin, S., & Genazzio, M. A. (2021). Validation of SMAP Soil Moisture at Terrestrial National Ecological Observatory Network (NEON) Sites Show Potential for Soil Moisture Retrieval in Forested Areas. IEEE Journal of Selected Topics in Applied Earth Observations and Remote Sensing (Vol. 14, pp. 10903–10918). https://doi.org/10.1109/jstars.2021.3121206

Colliander, A., Cosh, M. H., Kelly, V. R., Kraatz, S., Bourgeau-Chavez, L., Siqueira, P., A. Roy, A.G. Konings, N. Holtzman, S. Misra, D. Entekhabi, P. O'Neill, S.H. Yueh.: SMAP Detects Soil Moisture under Temperate Forest Canopies. Geophysical Research Letters. Vol. 47. https://doi.org/10.1029/2020GL089697, 2020.

---

## Author Comment (AC3)

The review comments are shown in black, and the author responses are in red.

We would like to thank the reviewer for reviewing our manuscript. Please find below our responses to the comments.

This study investigates the relationship between precipitation and soil moisture in Finland using satellite-based data from NASA's SMAP mission, supported by ground-based radar and in situ measurements. It applies the SM2RAIN algorithm to estimate precipitation from soil moisture and characterizes soil moisture dry down patterns by fitting exponential models. The analysis covers two years (2018–2019) and highlights spatial and temporal variability in soil moisture, as well as challenges posed by snowmelt and water bodies in SM retrievals. The findings are valuable for understanding Arctic hydrological processes and suggest potential for broader application across the high-latitude regions.

The study is scientifically sound, methodologically thorough, and clearly communicated. It effectively combines multiple data sets and provides insight into soil moisture dynamics in a challenging environment. The use of dry down modeling and SM2RAIN adds value, and the discussion acknowledges key limitations transparently. Here are comments for further improvement before publications.

The SM2RAIN algorithm is applied assuming that evapotranspiration and runoff are negligible during precipitation events. While this assumption is common, it may not always hold true in regions such as Finland, particularly in wetland areas or under light rainfall with significant canopy interception and latent heat fluxes. It would strengthen the study to assess the potential bias this simplification introduces, possibly by incorporating flux tower ET data or ERA5-Land evapotranspiration estimates.

First, we would like to clarify that the neglected runoff term only includes surface runoff, whereas subsurface runoff is included in the SM2RAIN algorithm. Specifically, the drainage term (Eq. 2 in the manuscript), contains the subsurface runoff. We will clarify this in the manuscript.

Brocca et al. (2015) studied the impact of various terms on the precipitation (P) estimates derived from the SM2RAIN algorithm. Their research demonstrated that soil moisture variations and the drainage term are the most significant contributors, accounting for over 90% of the simulated P estimates. The study concluded that neglecting evapotranspiration and surface runoff terms does not weaken the algorithm's performance. Additionally, the use of satellite data with relatively coarse resolution further reduces the impact of surface runoff. The portion that does not infiltrate—due to factors such as impervious land cover or soil—may re-infiltrate downstream within the scale of a SMAP grid cell (Brocca et al., 2019).

Nevertheless, we acknowledge that limitations exist in the SM2RAIN algorithm. In future studies, analyzing more complex versions of the SM2RAIN algorithm would be worthwhile, although that is beyond the scope of this study. As this is the first study to analyze SM2RAIN's performance in the Arctic and in regions with seasonally frozen ground, we consider that the current simple form of the SM2RAIN algorithm is adequate. Now that the performance has been established in this study, the next step would be to study more complex versions of SM2RAIN and its performance in future studies. We will add discussion about the limitations and future perspectives in the manuscript.

The use of fixed empirical parameters (Z, a, b) from Brocca et al. (2014) across all grid cells is practical but may not adequately capture the spatial variability in soil hydraulic behavior. Given the known heterogeneity in Finnish soils, vegetation, and hydrological settings, a spatially stratified parameterization based on land cover or soil texture (e.g., from SoilGrids or FAO datasets) would be more representative. Alternatively, local calibration using available in situ or radar-based precipitation data could be tested in a subset of locations to explore the potential for improved accuracy.

We will calibrate the parameters and will take into account land cover or soil texture as suggested.

The exponential decay model used for drydowns is based on a first-order assumption that may oversimplify soil water loss processes in certain conditions, especially where rooting depth, vegetation type, or soil layering affects moisture dynamics. Exploring alternative drydown functions—such as bi-exponential or piecewise linear decay—on a subset of events could help determine whether the exponential model remains robust across regimes.

We acknowledge that limitations exist in the exponential decay model used for drydowns. In future studies, analyzing alternative drydown functions would be worthwhile, although that is beyond the scope of this study. As this is the first satellite-based study to analyze drydowns in the Arctic, we consider that using only the exponential decay model in this study is adequate. Now that the performance has been established in this study, the next step would be to study alternative drydown functions in future studies. We will add discussion about the limitations and future perspectives in the manuscript.

The study uses the SMAP L3 enhanced product oversampled to a 9 km grid, though the native resolution remains approximately 33 km. This discrepancy could lead to spatial artifacts or smoothing, especially near land-water boundaries or heterogeneous terrain. Clarification is needed on whether any additional spatial processing or filtering was applied to the SMAP data prior to analysis. It would also be useful to discuss the potential influence of footprint overlap on statistical independence in the drydown fitting.

We have not applied any additional spatial processing to the SMAP data. However, the radar data has been resampled to match the 33 km SMAP footprint. We will edit the text to clarify this and add discussion as suggested.

Only the descending (6 a.m.) SMAP overpasses are used, which is justified in terms of stability and retrieval quality. However, excluding ascending passes may overlook sub-daily variability—particularly relevant in summer months when diurnal SM cycles can be significant. The authors are encouraged to briefly quantify potential information loss from this exclusion or justify it further using SMAP data quality metrics for ascending overpasses.

It is true that some information may be lost due to using only the descending overpasses. However, the minor but systematic biases between the ascending and descending retrievals (e.g., Colliander et al., 2022) justify using only the descending retrievals for increased stability, consistency, and retrieval quality to characterize the drydowns. Combining the ascending and descending overpasses would introduce additional noise in the combined timeseries, degrading the stability of the timeseries, resulting in less accurate analysis. We will clarify this in the manuscript.

The approach of averaging in situ soil moisture within SMAP footprints is appropriate, but spatial heterogeneity within grid cells—especially in complex environments such as fens versus forests—could introduce representativeness errors in the validation. A quantitative assessment of subgrid variability (e.g., standard deviation or interquartile range across in situ probes) and its impact on satellite comparison would strengthen the validation component of the study.

We have only used averaged in situ measurements in Sodankylä, where multiple in situ measurements are available. In Hyytiälä, two in situ measurements are available, and we did not apply any averaging but showed both measurement sites individually (Fig. 6b in the manuscript). For other locations, only one measurement is available, so no averaging was made. However, we acknowledge that spatial variability in SM within a grid cell can be considerable. This is evident for example in Fig. 6b, where the two in situ measurements show large discrepancies. Therefore, we will analyze the standard deviation for Sodankylä in situ measurements and add discussion in the text accordingly.

Surface classification based on SMAP retrieval flags is a useful step; however, the interpretation of SMmin and τ values could be improved with more granular land surface information. Adding categorical breakdowns by vegetation type or soil class could help explain observed spatial patterns and potential retrieval biases, particularly in regions affected by water bodies. Additionally, it would be helpful to explicitly state the proportion of SMAP observations excluded by quality flags and assess whether this introduces systematic seasonal or regional sampling bias—e.g., due to frozen conditions or persistent cloud/snow cover.

We will add more detailed analysis of $SM_{min}$ and τ values based on soil classes as suggested, and will also quantify the proportion of SMAP observations excluded by quality flags.

The springtime SM–P relationship is notably weaker due to snowmelt contributions to SM increases. While this is acknowledged, a more detailed discussion or methodology for separating snowmelt-induced SM changes from rainfall-driven changes would be beneficial. This could involve the use of modeled snowmelt fluxes from ERA5-Land, or analysis of coincident snow cover and air temperature data.

We will add discussion on this topic as suggested.

The study is limited to two years of data (2018–2019), which are suitable for identifying seasonal patterns but may not capture interannual variability. It would be useful to briefly contextualize whether either year exhibited anomalous precipitation or temperature relative to the climatological baseline or mention how a longer time series could help address questions of variability or trend detection.

We will edit the text according to the comment.

Given the stated potential for Arctic-wide application, the manuscript could benefit from a short discussion on the method's suitability for permafrost-affected regions. Permafrost soils introduce unique SM dynamics, including seasonal freeze-thaw cycles, active layer constraints, and limited drainage. Whether this methodology could be adapted or tested in northern permafrost zones (e.g., in northernmost Finland or elsewhere) is worth exploring.

We will add a brief discussion on this topic to the manuscript.

The use of the term "saturation" to describe SMAP retrievals near water bodies may be misleading for readers unfamiliar with remote sensing. Clarifying whether this refers to sensor saturation, retrieval floor/ceiling effects, or actual physical saturation of the soil column would improve clarity.

We will clarify this in the text.

References:

Brocca, L., Massari, C., Ciabatta, L., Moramarco, T., Penna, D., Zuecco, G., Pianezzola, L., Borga, M., Matgen, P. and Martínez-Fernández, J.: Rainfall estimation from in situ soil moisture observations at several sites in Europe: an evaluation of the SM2RAIN algorithm. Journal of Hydrology and Hydromechanics, (3), 201-209, https://doi.org/10.1515/johh-2015-0016, 2015.

Brocca, L., Filippucci, P., Hahn, S., Ciabatta, L., Massari, C., Camici, S., Schüller, L., Bojkov, B., and Wagner, W.: SM2RAIN‑ASCAT (2007‑2018): global daily satellite rainfall data from ASCAT soil moisture observations, Earth Syst. Sci. Data, 11, 1583‑1601, https://doi.org/10.5194/essd-11-1583-2019, 2019.

Colliander, A., Reichle, R. H., Crow, W. T., Cosh, M. H., Chen, F., Chan, S., ... and Yueh, S. H.: Validation of soil moisture data products from the NASA SMAP mission. IEEE Journal of selected topics in applied earth observations and remote sensing, 15, 364-392, https://doi.org/10.1109/JSTARS.2021.3124743, 2022

---

## Author Response (AR2)

The review comments are shown in black, the author responses are in red, and text from the revised manuscript in *red italics*.

**Reviewer #1**

Many thanks for addressing most of my suggestions. There is one point though I am quite adamant to see corrected in the published version about the spatial resolution (my "line 112" comment). I did not give the right reference then and should have put SMAP Handbook, but in this document the spatial resolution is well detailed on p 35 and is given as 39x47 km which is not 36 (again this is not the spatial resolution but the spatial sampling).

We would like to thank the reviewer for reviewing our manuscript. We have edited the text according to the comment as follows (lines 121-123):

*SMAP has an effective field of view of 39 km × 47 km (Entekhabi et al., 2014). The Level 3 soil moisture products are gridded to a 36-km fixed Earth grid and oversampled into a 9 km grid (Entekhabi et al., 2014; O'Neill et al., 2021a).*

**Reviewer #2**

We have edited our previous responses based on the editor's comment to provide specific explanations of the revisions we have made.

Major comments:

Lines 179-172 "Using…. 2014)": Not clear. Hence the SM2RAIN algorithm was not calibrated? This approach is not correct: SM2RAIN parameters need to be calibrated to each pixels, as they are dependent from the soil characteristic of the pixels. All the works of Brocca et al. after 2013 relied on this, to overcome the limits of 2013 paper. SM2RAIN should be calibrated against observations (without considering SM data obtained in frozen conditions). If the Authors prefer to not calibrate the data, they should still use some regionalization procedure to obtain SM2RAIN weight for each pixel (e.g. Filippucci et al 2021; 2022, here applied to ASCAT or S1 data). The parameters can change greatly according to the area. Moreover, this approach do not consider SM2RAIN temporal filter T. This would be important to assess the real performance of SM2RAIN in the area.

We have revised the approach and included both calibration and filtering as suggested. The SMAP data were temporally filtered using the exponential filter approach. The parameters Z, a, and b were calibrated by minimizing the root-mean-square error between the simulated and radar-based 5-day accumulated rainfall. We used data from year 2018 for calibration and data from year 2019 for validation. The revised text is below the next comment.

Moreover, SM2RAIN use interpolated daily data, normalized between 0 and 1. Did the Authors applied interpolation and normalization? It should be written and, if not applies, please explain why.

Yes, we have applied both interpolation and normalization. In case of missing SMAP data, the data were linearly interpolated, and a maximum data gap of three days was considered for the interpolation.

We have revised the text to clarify the preprocessing and calibration steps (lines 231-238):

*Before using SMAP SM observations as input for the algorithm, the data needed to undergo a few preprocessing steps. First, in case of missing SMAP data, linear interpolation was used to generate daily values, with a maximum allowable data gap of three days for interpolation. Second, the SMAP data were*

*rescaled between 0 and 1, as relative SM is used as input in the SM2RAIN algorithm. Third, to address high-frequency fluctuations in satellite-based SM observations caused by measurement and retrieval errors, the SMAP data were temporally filtered using the exponential filter approach (Wagner et al., 1999). After these preprocessing steps, the SMAP data were ready to be applied to the SM2RAIN algorithm. The parameters Z,*
45    *a, and b were calibrated by minimizing the root-mean-square error between the simulated and radar-based 5-day accumulated rainfall. We used data from year 2018 for calibration and data from year 2019 for validation.*

Regarding SM2RAIn quality, the error introduced by the snowmelt was already analyzed in Brocca et al., 2019, which applied mask to sm data during melting (0<T<3) in order to avoid this kind of noises. This should be implemented also here, to assess the very capabilities of NASA SMAP data.

50    We have revised the approach and excluded the melt season by only including Jun-Sep in the SM2RAIN analysis. We have edited the text (lines 238-240):

*Since snowmelt affects SM and may lead to erroneous P estimates, we excluded the snowmelt season by only including the months from June to September in the SM2RAIN analysis.*

Line 419 "while dense….challenge": I don't think that this conclusion is supported by this analysis. Dense
55    vegetation show a realistic pattern, but it is not verified that the SM beneath the vegetation is actually sensed. One way to obtain a confirmation would be to use SM2RAIN to evaluate precipitation time series for all the pixels and then assess its performance per pixel (with indices as RMSE or Pearson correlation or BIAS). If the results in the densely vegetated area are similar to those around, then this indicates that the capacity of SMAP to estimate SM under vegetation are good. But I expect to find worse correlation. In general, I understand that
60    the lack of SM data allow to validate SM just in few areas, but precipitation data are available for all the study areas, hence I encourage to use SM2RAIn to gain more information regarding the SM data quality

It is true that the lack of reference SM data complicates the validation of satellite-based SM retrievals. However, research has shown that SMAP is able to detect soil moisture beneath the vegetation (Colliander et al., 2020; Ayres et al., 2021). Using P data together with the SM2RAIN algorithm to evaluate SMAP SM
65    retrievals would be a worthwhile study, but we consider it being out of scope of this study. We have edited the text as follows (lines 502-509):

*Grid cells containing water bodies (classified as "water" and "water and dense vegetation") show a mode in $SM^*_{min}$ at 0.02 $m^3$ $m^{-3}$, alongside a lower and less distinct peak at higher values. Since 0.02 $m^3$ $m^{-3}$ is the lower limit for SMAP SM retrievals, this may introduce an artifact in the results. Conversely, grid cells without water*
70    *bodies ("low vegetation" and "dense vegetation") display more consistent distributions, each with a single peak around 0.13 $m^3$ $m^{-3}$. These differences in distribution suggest that surface conditions significantly influence the SMAP SM retrievals. Specifically, this result indicates that water bodies complicate SM retrieval, while dense vegetation alone does not pose as much of a challenge. This conclusion is further supported by previous findings demonstrating SMAP's capability to detect SM beneath vegetation cover (Ayres et al., 2021;*
75    *Colliander et al., 2020).*

Minor comments:

Lines 55-57 "While these….2019)": This is not completely true, as, for example, brocca et al. do not exclude artic region, just periods in which the soil is frozen. Check and correct

We have edited the text as follows (lines 62-67):

80    *While these studies highlight the potential of this approach, they often exclude the Arctic region due to the challenges associated with SM retrieval (Ciabatta et al., 2018; Brocca et al., 2019; Koster et al., 2016; Zhang et al., 2019). Brocca et al. (2019) included the Arctic in their SM2RAIN analysis using data from the Advanced*

*Scatterometer (ASCAT). However, research has shown that the Soil Moisture Active Passive (SMAP) mission outperforms ASCAT in high-latitude environments (Chen et al., 2018; Zhang et al., 2020), highlighting the importance of evaluating whether SMAP can yield more accurate P estimates in these regions.*

Lines 267-269 "This….site": Indeed, the discrepancies in SM between the two sites could be related to different precipitation pattern in the large SMAP pixel. However, it could be also related to error in SM measurement. Since Radar data have 250 meters spatial resolution, you could compare the radar Precipitation obtained in the pixels nearest to the Hyytiälä stations, to ensure that they are correlated with it and therefore there is no error in the observed SM measurements.

Thank you for the suggestion. Rather than using radar data from the pixel nearest to the Hyytiälä stations, we decided to use the in situ P measurements from all the locations shown in Fig. 2a. We have added the in situ P data in Figs. 6 and S3 and revised the text accordingly (lines 312-338).

Line 363 "Figure…. on SM": Not clear, please expand the reasoning behind the sentence

There is a typo in the sentence, and it should read Figure 11 instead of Figure 10. We have corrected the typo and also edited the sentence to make it clear for the reader (lines 432-433):

*Figure 11 indicates that SM\*$_{min}$ is particularly influenced by surface classification, as its spatial variability closely resembles those shown in the surface flag map (Fig. 2c).*

Lines 386-387 "Our analysis….Fig. 7),": To be checked after proper SM2RAIN calibration (see major comment #1)

We have reviewed the conclusions after applying the calibration and preprocessing steps and revised the text accordingly. Although some minor differences were observed, the main conclusions remained unchanged.

References:

Ayres, E., Colliander, A., Cosh, M. H., Roberti, J. A., Simkin, S., & Genazzio, M. A. (2021). Validation of SMAP Soil Moisture at Terrestrial National Ecological Observatory Network (NEON) Sites Show Potential for Soil Moisture Retrieval in Forested Areas. IEEE Journal of Selected Topics in Applied Earth Observations and Remote Sensing (Vol. 14, pp. 10903–10918). https://doi.org/10.1109/jstars.2021.3121206

Colliander, A., Cosh, M. H., Kelly, V. R., Kraatz, S., Bourgeau-Chavez, L., Siqueira, P., A. Roy, A.G. Konings, N. Holtzman, S. Misra, D. Entekhabi, P. O'Neill, S.H. Yueh.: SMAP Detects Soil Moisture under Temperate Forest Canopies. Geophysical Research Letters. Vol. 47. https://doi.org/10.1029/2020GL089697, 2020b.